# A coordinated transcriptional switching network mediates antigenic variation of human malaria parasites

Xu Zhang[1], Francesca Florini[1], Joseph E Visone[1], Irina Lionardi[2], Mackensie R Gross[1], Valay Patel[1], Kirk W Deitsch[1]*

[1]Department of Microbiology and Immunology, Weill Cornell Medical College, New York, United States; [2]Jill Roberts Center for Inflammatory Bowel Disease, Weill Cornell Medical College, New York, United States

**Abstract** Malaria parasites avoid immune clearance through their ability to systematically alter antigens exposed on the surface of infected red blood cells. This is accomplished by tightly regulated transcriptional control of individual members of a large, multicopy gene family called *var* and is the key to both the virulence and chronic nature of malaria infections. Expression of *var* genes is mutually exclusive and controlled epigenetically, however how large populations of parasites coordinate *var* gene switching to avoid premature exposure of the antigenic repertoire is unknown. Here, we provide evidence for a transcriptional network anchored by a universally conserved gene called *var2csa* that coordinates the switching process. We describe a structured switching bias that shifts overtime and could shape the pattern of *var* expression over the course of a lengthy infection. Our results provide an explanation for a previously mysterious aspect of malaria infections and shed light on how parasites possessing a relatively small repertoire of variant antigen-encoding genes can coordinate switching events to limit antigen exposure, thereby maintaining chronic infections.

**\*For correspondence:**
kwd2001@med.cornell.edu

**Competing interest:** The authors declare that no competing interests exist.

## Editor's evaluation

This is an important study addressing the mechanisms of variant gene expression and switching in the malaria parasite *Plasmodium falciparum*. The work provides solid evidence supporting the existence of a non-random, highly structured switch pathway for var genes and identifies one var gene, called var2csa, as a sink node in this network. The findings consolidate previous observations and further the understanding of the enigmatic mechanism that underlies the regulation of the var gene family and antigenic variation in *P. falciparum*, which is paramount for immune evasion and acquired immunity and further influences malaria pathology.

## Introduction

Malaria is a disease of enormous historical significance that continues to exert a heavy burden on health and economic development in many regions of the world (*World Health Organization, 2021*). The most virulent of the human malaria parasites is *Plasmodium falciparum*, a mosquito-borne pathogen that infects the circulating red blood cells (RBCs) of its hosts. While resident within the RBC, the parasites modify the host cell cytoskeleton through the insertion into the RBC membrane of an adhesive protein called *Plasmodium falciparum* erythrocyte member protein 1 (PfEMP1) (*Baruch et al., 1995*; *Smith et al., 1995*; *Su et al., 1995*). This enables adhesion to the vascular endothelium (*Gardner et al., 1996*), thereby avoiding filtration and destruction in the spleen (*David et al., 1983*; *Barnwell et al., 1983*), however this also exposes PfEMP1 as a target for adaptive immunity (*Bull et al., 1998*;

**eLife digest** Malaria causes severe illness and deaths in hundreds of thousands of people each year. Most of them are young children in Sub-Saharan Africa. The disease is transmitted when a mosquito carrying single-celled *Plasmodium* parasites bites a human, introducing the parasites into the bloodstream, where they enter red blood cells.

When a red blood cell becomes infected, the parasite presents a protein on the cell's surface that the immune system can recognize to start fighting the infection. Immune cells then produce antibodies that flag infected cells for destruction, relieving the symptoms of the disease. To avoid being destroyed in this manner, the parasites repeatedly 'change' the protein that ends up on the surface of the red blood cells. With each change, the number of parasites rebounds, symptoms return, and the immune system must produce new antibodies. As the parasites and immune system battle it out, patients may experience repeated flare-ups of symptoms for well over a year.

To change the protein that is presented on the surface of red blood cells, *Plasmodium* parasites switch the genes in the *var* gene family on and off one at a time. Each of these genes encodes a different surface protein, and the parasites may cycle through the entire *var* gene family during an infection. However, it remains a mystery how the millions of infecting parasites coordinate to produce the same surface protein each time.

Zhang et al. show that a gene from *Plasmodium* parasites called *var2csa* is responsible for coordinating protein switching through a set pattern that allows the parasites to synchronize which protein they switch to next. Deleting the *var2csa* gene in malaria parasites blocks protein switching and disables this coordinated immune evasion tactic.

Zhang et al.'s experiments provide new insights about protein switching in malaria parasites. Further research may help scientists characterize each step in the process and identify which steps can be targeted to treat malaria. While not a cure, treatments that disable protein switching could reduce the number of times patients relapse and relieve symptoms. More generally, the results of Zhang et al. describe a mechanism for coordinated gene expression that may be used in organisms other than *Plasmodium*, including humans.

---

*Giha et al., 2000*). To escape clearance by antibodies that recognize PfEMP1, parasites systematically change the form of PfEMP1 they express, thus continuously altering their antigenic signature and promoting a chronic infection that can extend over a year (reviewed in *Deitsch and Dzikowski, 2017*).

Different forms of PfEMP1 are encoded by individual members of the *var* gene family, a collection of highly variable, paralogous genes numbering between 45 and 90 and distributed throughout the parasite's genome, primarily within subtelomeric regions or as tandemly arranged clusters within the interior of the chromosomes (*Otto et al., 2018*). Epigenetic mechanisms ensure that only a single gene is actively transcribed at a time (*Scherf et al., 1998*; *Deitsch et al., 1999*), a process with parallels to allelic exclusion of immunoglobulin genes (*Corcoran, 2005*) or mutually exclusive expression within the olfactory receptor gene family in mammals (*Rodriguez, 2013*). However, unlike these systems in which selection of the active gene is part of the differentiation process and thus permanent, *var* gene activation and silencing are reversible, thereby enabling parasites to switch which *var* gene is expressed, thus changing the form of PfEMP1 on the infected cell surface. Many attributes of the epigenetic processes that control *var* gene expression, including the histone modifications that mark a gene for activation or silencing, are shared with model eukaryotes (*Chookajorn et al., 2007*; *Lopez-Rubio et al., 2009*; *Flueck et al., 2009*). However, the added complexity of switching which gene is active while maintaining mutually exclusive expression is largely without precedent outside of pathogenic organisms. Moreover, given that the number of *var* genes within the parasite's genome is relatively limited, switching events must be coordinated to avoid rapid exhaustion of the parasite's variant capacity. Specifically, uncoordinated, random *var* gene switching by individual parasites within a circulating population that can number in the billions would result in rapid exposure of the entire *var* repertoire. In contrast, *P. falciparum* infections are characterized by rising and falling waves of parasitemia, with each wave representing a population of parasites expressing a single or small number of *var* genes (*Bachmann et al., 2011*; *Bachmann et al., 2019*; *Kaestli et al., 2004*). Modeling studies have suggested that in semi-immune hosts, an underlying structure or coordination

to the switching process would be optimal for extending an infection (*Recker et al., 2011*; *Noble and Recker, 2012*; *Noble et al., 2013*) by limiting activation to a single or small number of genes at a time within the circulating parasite population. However, while communication between parasites has been suggested (*Mantel et al., 2013*; *Regev-Rudzki et al., 2013*), there is no evidence that this influences *var* gene expression patterns and there does not appear to be a strict switching order within the *var* gene family, therefore how this is accomplished remains entirely without explanation.

In 2011, Recker et al. explored this phenomenon using mathematical modeling based on data from switching events observed in cultured parasites (*Recker et al., 2011*). They proposed that parasites do not switch directly from one gene to another, but rather that a population of parasites switches from a dominant *var* gene, through a switch-intermediate state in which many genes are transiently expressed, to either a new dominant transcript or back to the original. This model also predicted that specific genes could serve as either 'source' or 'sink' nodes and provide additional structure to the network. The model provided a compelling explanation for how a population of parasites that can reach up to $10^{10}$ individuals can display seemingly coordinated switching patterns without the need for communication between cells or a fixed order of gene activation. Further, it described both how parasites can extend a chronic infection as well as successfully reinfect individuals when partial immunity exists. However, to date no experimental data have provided a molecular basis for such a structured network, the hypothetical switch-intermediate state or the source/sink nodes predicted by this model.

Here, we describe molecular genetic evidence that fulfils many of the predictions of the previously proposed mathematical model. We identified a universally conserved, unique *var* gene that displays characteristics of a sink node and plays a key role in coordinating transcriptional switching events. Deletion of this locus disrupts *var* gene switching, resulting in parasites that have a drastically reduced ability to change *var* gene expression, thus disabling the process of antigenic variation that is required to maintain a chronic infection. In addition, we describe an underlying pattern of biased *var* transcriptional activation which shifts overtime, providing the potential for a coordinated pattern of expression switching that could prevent premature exposure of the parasite's variant repertoire over the length of an infection. These data extend our understanding of antigenic variation beyond mathematical models and represent an important step forward in understanding the pathogenic nature of human malaria.

## Results

### Clonal parasite lines display either 'single' or 'many' *var* gene expression profiles and can transition between these two states

The mathematical model derived by Recker et al. was based on observations of *var* expression patterns observed in recently isolated clonal parasite populations (*Recker et al., 2011*). The proposed transition from expression of a single *var* gene, through an intermediate state in which many genes are expressed, then back to a single gene that becomes dominant in the population is referred to as the 'single-many-single' model for *var* gene switching (*Figure 1A*). This is similar to the two-step model for olfactory receptor gene activation in vertebrates, which has recently been shown to involve an initial, transient state in which large numbers of genes are expressed at a low level, followed by selection of a single gene for stable expression (*Tan et al., 2015*). Given the observations from this model system and previous reports of *var* gene switching patterns, we attempted to experimentally validate the SMS model of structured switching and to determine the molecular requirements that underlie the hypothetical *var* gene network that forms the foundation for this pathway.

The original proposal of the SMS model was based on analysis of *var* gene expression by Northern blotting and quantitative real-time RT-PCR (qRT-PCR) using the IT and 3D7 genetic backgrounds (*Recker et al., 2011*). Since this original description, the method for determining *var* expression patterns has become widely used for NF54 (*Delemarre and van der Kaay, 1979*) and 3D7 (*Walliker et al., 1987*) through the further refinement of a standardized qRT-PCR method developed specifically for this genetic background (*Salanti et al., 2003*). This standardized method enables rapid, quantitative assessment of the expression level of each *var* gene in the parasite's genome. We employed this method to examine *var* gene expression profiles obtained from recently cloned parasite lines derived from a single parent population of 3D7. Specifically, we determined the expression level of each *var* gene within each subcloned population approximately 5 weeks after cloning, the earliest time point

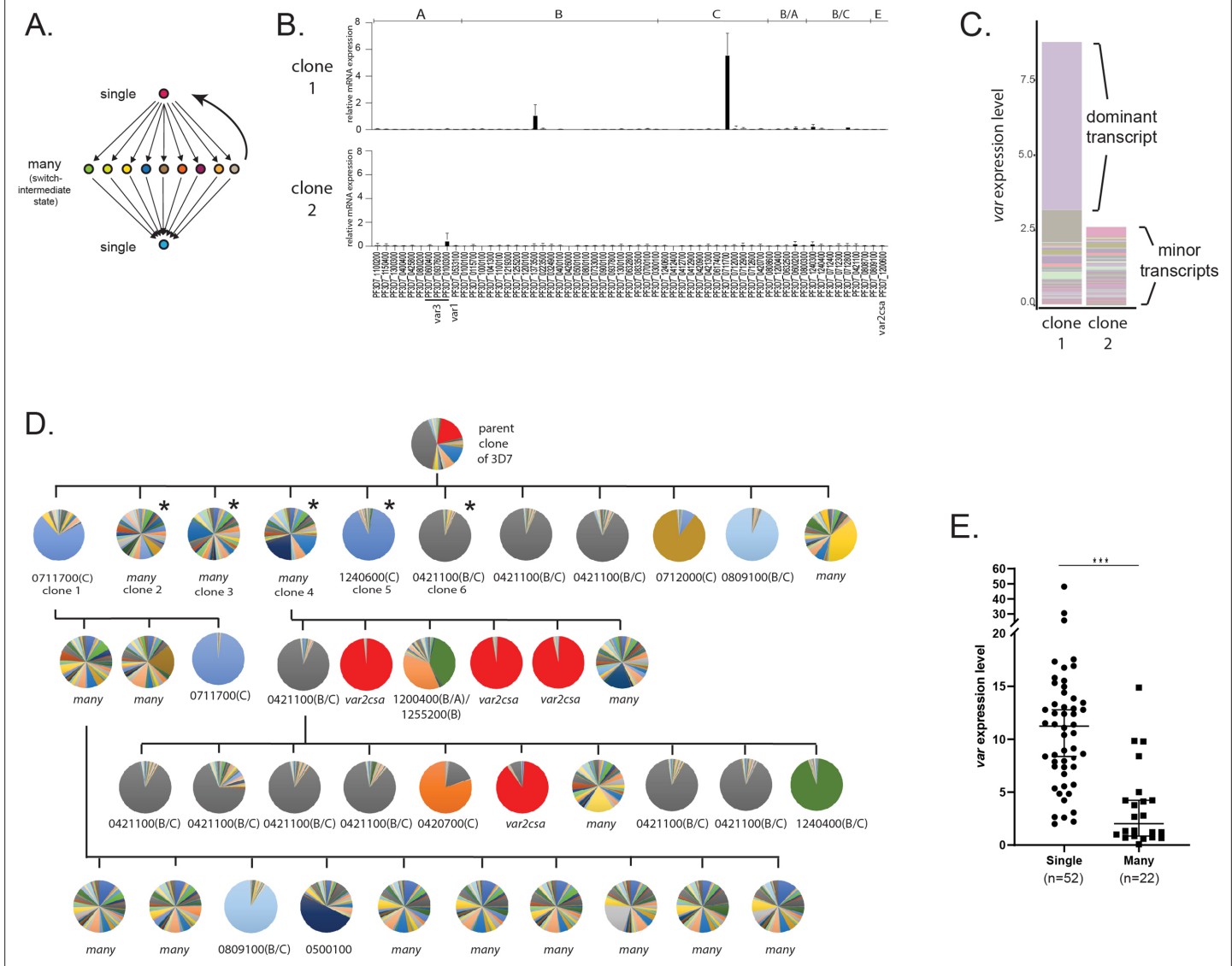

**Figure 1.** Detection of 'single' and 'many' *var* expression profiles in cultured parasite populations. (**A**) Schematic representation of the single-many-single (SMS) model for *var* gene transcriptional switching. Each circle represents expression of an individual *var* gene and arrows represent switches in expression. Switching events are hypothesized to transition from activation of a single gene (top) to a broad range of genes (middle) to a different single gene (bottom) or back to the original gene (reverse arrow). (**B**) *var* gene expression profiles for two clonal populations that display the single phenotype (clone 1, top) or the many phenotype (clone 2, bottom). *var* transcription levels were determined using a standardized quantitative real-time RT-PCR (qRT-PCR) protocol with the expression of each individual *var* gene displayed as relative copy number in the histogram. Error bars represent standard deviation of three biological replicates. (**C**) Total *var* expression for the two subclones shown in B, with transcripts from each individual *var* gene shown in a different color. For clone 1, transcripts from the dominant gene are marked, while both clones display similar levels of minor transcripts. (**D**) Clone tree of wildtype 3D7 parasites. Pie charts display the *var* expression profile for each subcloned population with each slice of the pie representing the expression level of a single *var* gene. Vertical and horizontal lines delineate sequential rounds of subcloning by limiting dilution. For parasite populations that display a dominantly expressed *var* gene, the annotation number is shown below the pie chart with the *var* type shown in parenthesis. The five subclones marked with an asterisk are further analysed in *Figure 2*. (**E**) Total *var* expression levels as determined by qRT-PCR for 74 subclones (see *Figure 1—figure supplement 1*). The median ± 95% confidence interval is shown, and an unpaired *t*-test indicates a ***p < 0.0001.

The online version of this article includes the following figure supplement(s) for figure 1:

**Figure supplement 1.** Clone tree of wildtype 3D7 parasites.

**Figure supplement 2.** Examination of *var* gene expression profiles of recent clones of NF54.

from which we could obtain suitable parasite numbers. Interestingly, these subcloned populations displayed two fundamentally different expression patterns. Clones displayed either dominant, relatively high-level expression of a single *var* gene with low-level expression of other members of the family, as exemplified by clone 1 (*Figure 1B*, top panel), or alternatively they lacked expression of a dominant gene and instead only displayed heterogeneous, low-level expression of a large portion of the *var* gene family, as shown by clone 2 (*Figure 1B*, bottom panel). Measurement of the total level of *var* gene expression from all members of the family detected significantly fewer *var* transcripts in the expression profile of clone 2 (*Figure 1C*), consistent with the lack of a dominantly expressed gene and indicating that overall *var* expression is lower in this population in addition to being more heterogenous. Given that the number of generations after cloning was identical for both populations, these differences in *var* gene expression profiles presumably reflect differences in the initial *var* expression state or in switching frequency. Similar, rather dramatic differences in total *var* expression levels have been previously described for clonal lines derived from both the NF54 and IT parasite isolates (*Merrick et al., 2015*; *Janes et al., 2011*), suggesting this phenomenon is typical of cultured parasites of different genetic backgrounds. The two phenotypes, either high-level, stable expression of a dominant *var* gene or highly diverse, low-level expression of many genes, are consistent with the 'single' and 'many' expression states proposed in the SMS model (*Recker et al., 2011*).

If parasites transition between the single and many states as part of *var* expression switching as proposed in the SMS model, and if this model applies to the two different *var* expression profiles we observed in our recently cloned lines, then the phenotypes should be reversible. Specifically, it should be possible to obtain 'single' parasites from a population of parasites that cumulatively expresses 'many' *var* genes, and vice versa. To test this hypothesis, we used serial subcloning by limiting dilution to isolate and examine a large number of additional subclones (*Figure 1D*). As predicted, we again obtained parasite populations that either displayed low-level expression of many *var* genes or high-level, stable expression of one or a small number of genes, regardless of which phenotype was displayed by the population from which the clone was derived (*Figure 1D*; additional clones are shown in *Figure 2B*). Thus, it appears that parasites can transition between these two states, leading to populations that display heterogenous expression of many *var* genes or relatively stable expression of a single, dominant *var* gene. We also anticipated that populations primarily consisting of parasites in the 'many' state would display lower total *var* expression levels than populations that express one or two dominant *var* transcripts. To test this prediction, a larger collection of 74 recently subcloned lines were examined and each subclone was determined to be primarily in either the 'single' or 'many' state (see *Figure 1—figure supplement 1*). Specifically, populations in which over 50% of the total *var* expression profile was derived from one or two genes were defined as 'singles' while populations with more diverse expression patterns were considered 'manys'. When total *var* transcript levels were determined by qRT-PCR, levels were significantly lower in the 'many' lines (*Figure 1E*), consistent with the initial observations shown in *Figure 1B* and the SMS model. To ensure that this phenomenon was not a property unique to parasites that had been in continuous culture for decades, we repeated the subcloning and *var* expression analysis with a line of NF54 (the original isolate from which 3D7 was derived) that had not been in culture for as many replicative cycles and that maintains many characteristics of parasites recently isolated from the field, for example gametocytogenesis and knob formation. These subcloned lines similarly displayed either the 'single' or 'many' phenotypes, with the 'singles' displaying higher levels of total *var* transcripts (*Figure 1—figure supplement 2*). Note that there is a very clear and statistically significant difference in total *var* expression levels between populations categorized as 'single' or 'many', however there is some overlap. We hypothesize this is because no population is completely homogeneous and that parasites in the single state contribute disproportionately to the overall expression profile of a population due to their higher expression level. Thus, mixed populations will display intermediate expression levels. Nonetheless the trend is very clear and statistically significant.

## Distinct semi-conserved pattern of *var* gene transcription underlies the 'single' and 'many' states and shifts slowly over time

In our *var* expression analysis of recently cloned parasites populations, we observed that populations in both the 'many' and 'single' states expressed a subset of the *var* gene family at a low level (*Figure 1C*). Importantly, the entire *var* gene family is not equally represented within the subset of

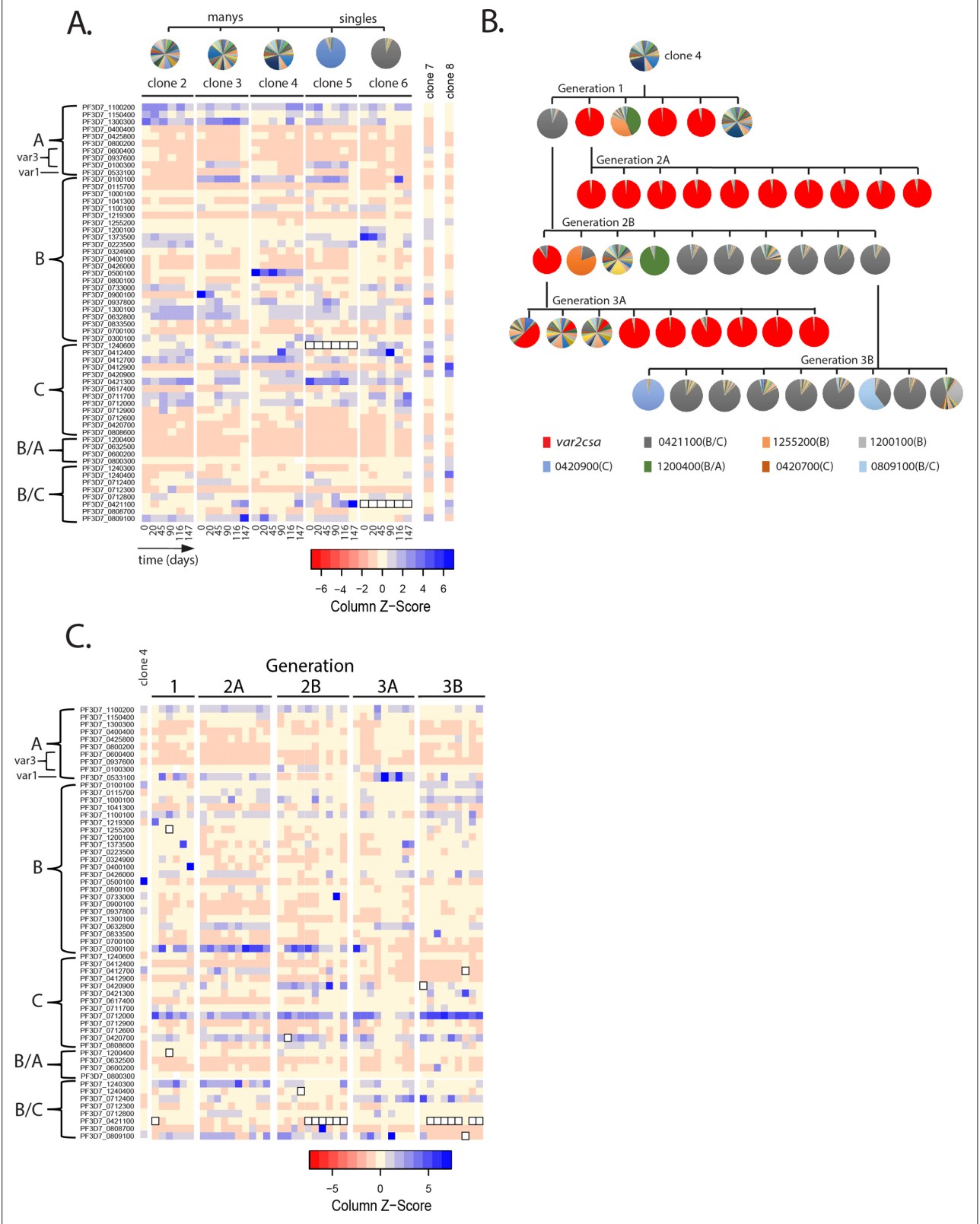

**Figure 2.** Detection of *var* gene minor transcript expression profiles in clonal populations overtime. (**A**) Heatmap of *var* minor transcripts for clones 2–6. The clones were all derived from the same parent population as shown in *Figure 1D*. Pie charts show the initial *var* gene expression profiles above the heatmap. The annotation numbers for each *var* gene are shown to the left of the heatmap and organized according to *var* type (left, *var2csa* is considered separately in *Figure 3*). Six time points are included for each clone. Two genetically identical parasite populations (clones 7

*Figure 2 continued on next page*

*Figure 2 continued*

and 8) also originally derived from 3D7 but grown separately for several years are shown for comparison. For clones 5 and 6, the dominant transcript (PF3D7_1240600 and PF3D7_0421100, respectively, marked by black boxes in the heatmap) was removed from the analysis to enable visualization of the minor transcripts. (**B**) Clone tree of wildtype 3D7. The tree is organized into three 'subclone generations' derived from initial clone 4 (top row). Pie charts display the *var* expression profile for each subclone. (**C**) Heatmap of *var* minor transcripts for the individual clones from each generation shown in B with the pattern of the parent population (clone 4) shown at the left for comparison. The annotation numbers for each *var* gene are shown to the left of the heatmap, and the gene order was organized according to *var* type. The order from left to right of each column in the heatmap corresponds to the order from left to right of the pie charts for each subclone generation shown in B. For parasites expressing the 'single' phenotype, the dominant transcript (marked by black boxes in the heatmap) was removed to enable visualization of the minor transcripts.

minor transcripts, suggesting that this low-level *var* gene activation is consistently biased toward certain *var* genes. If such intrinsic switching biases shift over time, they could shape the trajectory of *var* gene expression over the course of an infection and provide a model for how large populations of parasites could systematically cycle through their complement of *var* genes. This would lead to semi-coordinated *var* gene switching, thereby avoiding antibody-mediated clearance while protecting the majority of the variant repertoire. If true, this hypothesis provides a key insight into how antigenic variation with a small repertoire of genes can maintain a chronic infection, however direct evidence for shifting switching biases within the context of the SMS model has been lacking.

To specifically examine minor transcript expression, we compared the patterns of minor transcripts from five of the newly isolated clonal populations shown by asterisks in *Figure 1D* that displayed both the 'single' and 'many' expression profiles. RNA was obtained on days 0, 20, 45, 90, 116, and 147 after initiation of the experiment and the expression level of each *var* gene determined by qRT-PCR. The dominant *var* transcript was removed from the expression profile of 'single' populations and the expression levels of the remainder of the *var* gene family displayed in a heatmap, thus enabling easy visualization of the minor transcript expression pattern. *var2csa* is a unique *var* gene that often dominates cultured parasite populations and therefor was considered separately in *Figure 3*. We used this method to determine if a distinct pattern exists and if it shifts overtime (*Figure 2A*). The heatmap

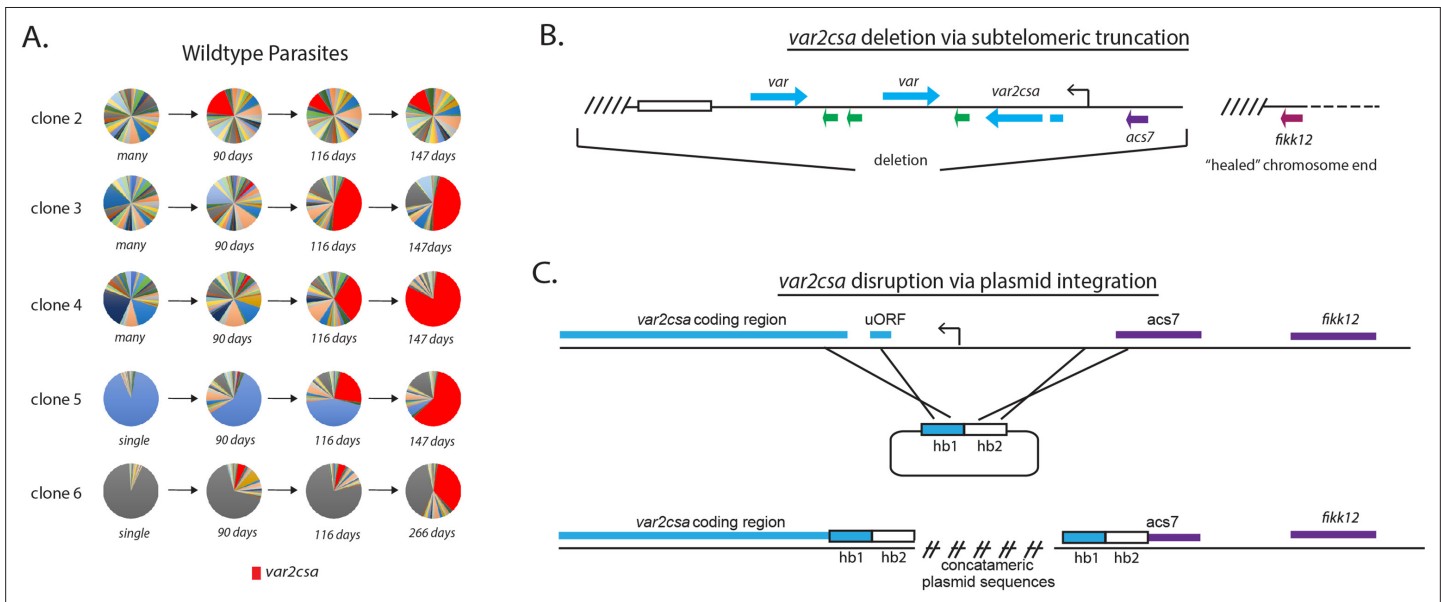

**Figure 3.** Convergence to *var2csa* expression in long-term cultures and targeted deletion of the *var2csa* locus. (**A**) Changes in *var* expression over time in five clonal parasite populations, three 'manys' (clones 2–4) and two 'singles' (clones 5 and 6), over several months of continuous culture. Expression of *var2csa* is shown in red. (**B**) Schematic diagram showing the truncation of the end of chromosome 12 by telomere healing. Telomere repeats are shown as slanted lines and the telomere-associated repeat elements (TAREs) are shown as an open box. The interior of the chromosome is displayed as a dashed line. A ~60 kb deletion, including three *var* genes (blue), three *rif* genes (green), and the *acs7* gene is shown. (**C**) Schematic diagram showing the *var2csa* locus and the plasmid containing homology blocks for targeted integration (hb1 and hb2). The crossed lines linking the plasmid to the chromosome signify sites of double cross over recombination leading to deletion of approximately 2.5 kb upstream of the *va2csa* gene, including the promoter.

shows that there are subsets of *var* genes that are more highly represented within the profiles of minor transcripts, and this pattern is largely shared within this group of closely related subclones. However, the pattern is not fixed and displays variation between subclones and over time. Two additional parasite populations that are genetically identical but had been cultured separately for several years (clones 7 and 8) displayed entirely different patterns of minor transcripts, providing additional evidence that the pattern is not fixed (*Figure 2A*). The similarity of the patterns displayed by the five subclones examined here likely reflects that these populations originated from a common original clonal population.

To further investigate the shifting pattern of *var* minor transcript expression, we employed a parallel approach. We again performed serial subcloning starting from clone 4, thereby obtaining additional closely related populations. If the pattern of minor transcripts is semi-conserved and shifts slowly over time, it should be largely shared in all the subcloned populations, regardless of whether they display a 'single' or 'many' *var* expression profile, or if they express different dominant *var* genes. Rather than following individual populations overtime, we instead isolated three 'subclone generations' (*Figure 2B*) and determined both the dominant *var* transcript as well as the pattern of minor *var* transcripts for each subcloned population. Similar to our previous observations, the subcloned populations displayed both 'single' and 'many' expression patterns, and the dominant *var* gene varied among the 'single' subclones (*Figure 2B*). Comparison of the *var* minor transcripts once again detected the existence of a distinct expression pattern that was semi-conserved but not fixed within this group of closely related populations (*Figure 2C*).

## Switching events converge overtime to *var2csa*, a conserved *var* locus

Previously, Mok et al. observed that after extended growth in culture, most parasite populations eventually converged to stable expression of a unique, highly conserved *var* gene called *var2csa* (UpsE), proposing that this gene could represent a default choice for *var* gene switching (*Mok et al., 2008*). We similarly previously observed convergence to *var2csa* expression when we artificially induced accelerated *var* gene switching by altering the activity of histone modifiers (*Ukaegbu et al., 2014*; *Ukaegbu et al., 2015*), further suggesting that the *var2csa* locus could occupy a unique position in the *var* switching hierarchy. To determine if *var2csa* likewise displayed preferential activation in our recently subcloned parasite populations, we more closely examined *var* transcript prevalence over time in the expression profiles for the five clonal lines shown in *Figure 2A*. Consistent with previous observations (*Mok et al., 2008*; *Ukaegbu et al., 2015*), all the clones, regardless of their initial switching frequency, eventually displayed significant expression levels of *var2csa* (*Figure 3A*). While how quickly each population converged to *var2csa* expression varied, in each instance, switching to *var2csa* was highly favored, suggesting that this *var* gene might occupy a unique position within the *var* gene switching hierarchy.

## The *var2csa* locus is required for efficient *var* gene switching

With respect to the SMS model, *var2csa* displays the properties of a sink node, and is by far the most likely *var* gene to become highly activated in our cultures (*Figure 3A*). This gene also displays several other properties that make it an atypical member of the family, including universal conservation extending to related species that infect chimpanzees and gorillas (*Zilversmit et al., 2013*; *Gross et al., 2021*) and the presence of a unique upstream regulatory region that includes an upstream open reading frame (uORF) resulting in translational repression of the mRNA (*Amulic et al., 2009*; *Bancells and Deitsch, 2013*; *Chan et al., 2017*). The gene can therefore transcribe mRNA without producing PfEMP1 and thus could be transcriptionally activated repeatedly over the course of an infection without generating an antibody response by the host. Based on these unusual properties, we hypothesized that *var2csa* represents the dominant sink node at the center of the SMS network. If correct, we predicted that loss of *var2csa* would significantly alter how parasites undergo *var* gene switching. We recently isolated numerous independent clonal parasite lines in which the entire subtelomeric region surrounding *var2csa* was deleted from the parasite's genome and the chromosome end 'healed' through de novo telomere addition (*Figure 3B*; *Zhang et al., 2019*). To complement this set of deletions, we employed CRISPR/Cas9-mediated genome editing to replace the *var2csa* upstream region with a plasmid construct that included a drug selectable marker, thereby deleting the entire promoter region and rendering the gene non-functional (*Figure 3C*). Several independent clones

were obtained from independent transfections of both 3D7 and NF54 that had this integration event and the structure of the resulting locus was verified by whole-genome sequencing. Examination of *var* expression profiles from each of these by qRT-PCR or RNAseq failed to detect any *var2csa* transcripts, indicating that transcriptional activity of the gene has been abolished. For all subsequent experiments, we observed no differences in phenotypes between the clones resulting from telomeric truncations and those derived from deletion of the *var2csa* promoter region.

To examine the effect of disruption of *var2csa* on *var* gene expression, *var* expression profiles were examined over an extended time in culture using qRT-PCR, and five representative clones (two derived from CRISPR-mediated disruption of the locus and three from subtelomeric deletion) are shown in *Figure 4A*. For all the clones, the initial *var* expression profiles resembled the two phenotypes previously observed when wildtype parasite populations were examined (i.e., either 'single' or 'many'). For example, three clonal lines (V2dis1, V2dis2, and ΔV2_3) displayed expression of a dominant *var* gene while two other clones (ΔV2_1 and ΔV2_2) displayed a heterogeneous mixture of numerous *var* transcripts (*Figure 4A*). These results suggest that the *var2csa* locus is not required for *var* gene activation or expression, since all the isolated clones were able to express *var* genes. To determine if *var2csa* has a role in *var* gene switching, these five clones were cultured continuously in parallel for various periods of time, ranging from 8 months to a year, monitoring *var* gene expression profiles periodically. Given that wildtype parasite populations tend to converge to expression of *var2csa* after extended time in culture (*Figure 3A* and *Mok et al., 2008*), we investigated what would happen in the absence of this locus. We observed that all the *var2csa*-deleted clones remained remarkably stable in their expression profiles, with the two 'single' lines maintaining dominant expression of the same active *var* gene throughout the experiment, a time period extending over a year (*Figure 4A*). Moreover, the two clones that displayed the 'many' state failed to converge to expression of any dominant *var* gene and continued to display highly heterogenous expression profiles (*Figure 4A*). These data indicate that convergent transcriptional activation is a unique property of *var2csa* and that no other *var* gene within the 3D7 genome can easily substitute upon *var2csa* deletion. In addition, the loss of *var2csa* appears to alter how efficiently parasites undergo *var* gene switching, resulting in parasites that, while some changes in expression are detectable, switch at a much slower rate. These results are consistent with a role for *var2csa* as the dominant sink node at the center of the SMS network.

## Loss of *var2csa* disrupts the underlying *var* gene transcriptional activation network

Given that the loss of *var2csa* altered *var* gene switching frequency (*Figure 4A*), we were interested in whether it might also influence the underlying pattern of minor *var* transcript expression that we observed in wildtype parasites (*Figure 2*). To investigate this, we generated a set of *var2csa*-deleted subclones that were all derived from the same parent clonal population but that displayed different *var* expression profiles (*Figure 4B*, top). Similar to our previous analysis, we then examined the minor transcript pattern over multiple time points of continuous culture (*Figure 4B*, heatmap). While patterns of expression are detectable within individual subclones, these patterns were much more diverse than displayed by the wildtype subclones and varied substantially between subclones and over time. To verify this conclusion in an unbiased, empirical fashion, we applied non-metric dimensional scaling based on Bray–Curtis dissimilarity (NMDS), treating the expression levels of each *var* minor transcript as variables (*Figure 4C*), and for comparison we included both wildtype and *var2csa*-deleted lines that had been similarly monitored for changes in minor transcript patterns over time. The patterns from all the wildtype clones closely clustered within a small region of the plot (shown in shades of light green), indicative of the similarity in the underlying expression patterns. In sharp contrast, the clusters from the different *var2csa*-deleted lines are scattered throughout the plot with only minor overlap, thus displaying a much-reduced stability of the pattern of *var* minor transcripts upon deletion of *var2csa* (*Figure 4C*). The increased variance within the clusters from the *var2csa*-deleted lines was confirmed by measuring the degree of sample dispersion, which can be inferred by the area on the plot (as defined by the mean distance of each time point from the group centroid) (*Figure 4D*).

Taken together, these data are consistent with the conclusion that the underlying pattern of *var* expression is semi-stable within closely related wildtype parasite populations but is less stable and displays much greater variability in the *var2csa*-deleted lines. If the patterns of minor transcripts analysed here reflect underlying on-switching rates for individual genes, a model emerges in which *var*

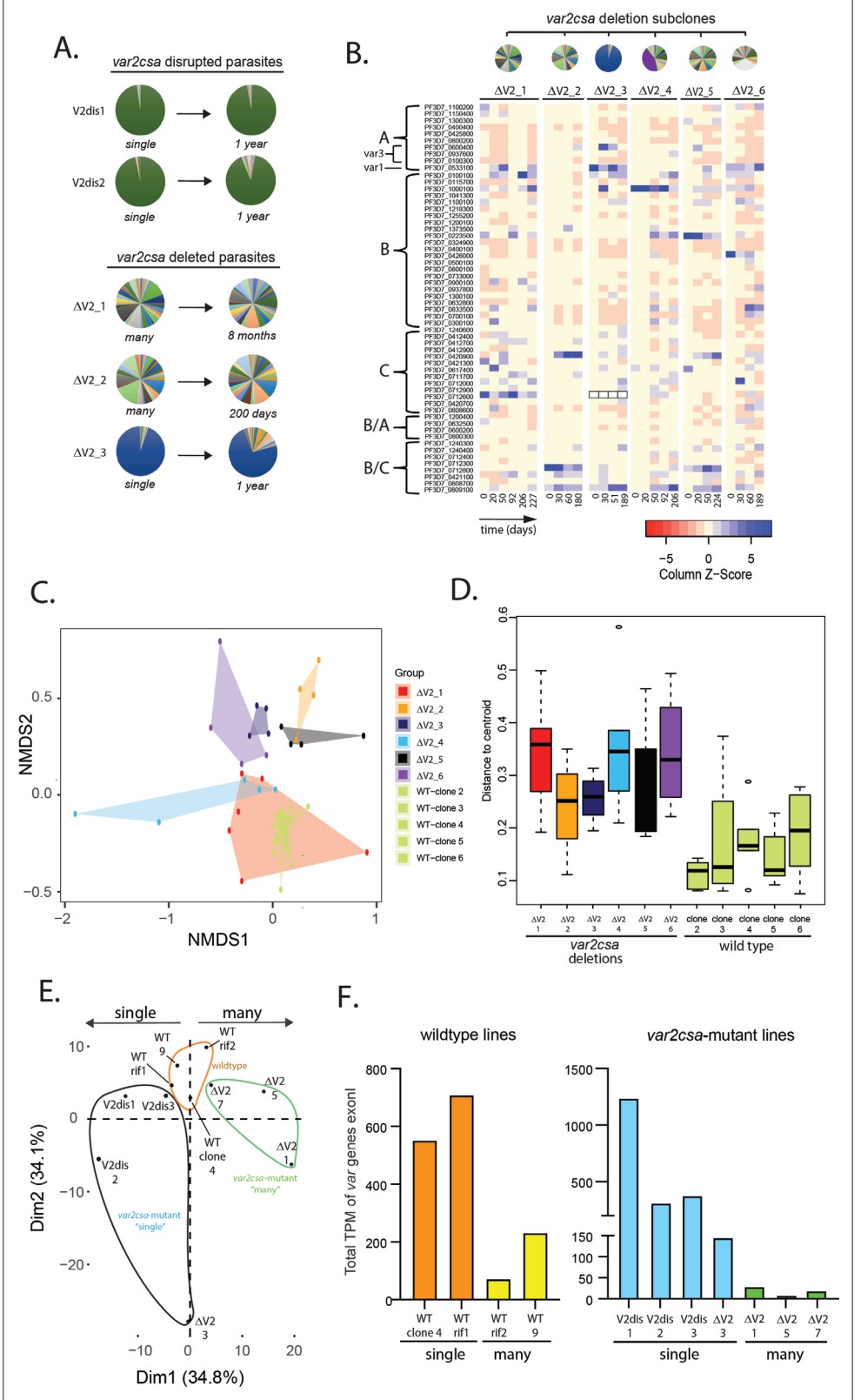

**Figure 4.** Alterations in *var* gene expression after deletion or disruption of *var2csa*. (**A**) Changes in *var* expression profile over time in five clonal parasite populations containing deletions or disruption of the *var2csa* locus, including two in which the locus was disrupted by plasmid integration (V2dis1 and V2dis2, top) and three in which the locus was deleted by chromosomal truncation (ΔV2_1, ΔV2_2, and ΔV2_3). V2dis1, V2dis2, and ΔV2_3 display

*Figure 4 continued on next page*

*Figure 4 continued*

a 'single' phenotype while ΔV2_1 and ΔV2_2 express many *var* genes. (**B**) Heatmap displaying the pattern of *var* minor transcripts over several time points for five *var2csa*-deleted lines. The clones were all derived from the same parent population, and pie charts showing the initial *var* gene expression profiles are shown above the heatmap. The annotation numbers for each *var* gene are shown to the left of the heatmap, organized according to *var* type. For ΔV2_3, the dominant transcript (Pf3D7_0712600, marked by black boxes in the heatmap) was removed to enable visualization of the minor transcripts. (**C**) Non-metric multidimensional scaling (NMDS) plot based on Bray–Curtis dissimilarities displaying the expression profiles of the *var* minor transcripts for the six *var2csa*-deleted lines alongside five wildtype lines. (**D**) The variability within each cluster shown in C is inferred by its area on the plot (as defined by the mean distance from the group centroid for each time point) and displayed in a boxplot. (**E**) Transcriptomic analysis of wildtype and *var2csa*-mutated parasite lines. Principal component analysis (PCA) is shown based on the normalized expression level of 5721 genes log10(fragments per kilobase of transcript per million mapped fragments (FPKM) + 1). RNAseq derived whole transcriptomes were obtained from 11 clonal parasite populations: two wildtype lines (WT 3.1 and WT 7) and two WT lines with a control plasmid integration into a *rif* gene (WT_rif1 and WT_rif2) are displayed in orange; four *var2csa*-mutated lines displaying a 'single' phenotype (V2dis1, V2dis2, V2dis3, and ΔV2_3) are displayed in blue; and three *var2csa*-mutated lines displaying a "many" phenotype (ΔV2_1, ΔV2_5, and ΔV2_7) are displayed in green. All analyses were performed using data obtained from synchronized populations ~16 hr after red cell invasion. (**F**) Comparisons of normalized *var* transcript counts from *var2csa*-mutated and wildtype lines displaying either the single (left) or many (right) phenotypes.

genes have differing propensities for activation, and that these change overtime, thereby enabling populations of parasites to systematically cycle through their repertoire of *var* genes over the course of an infection. The greater variability displayed by the *var2csa*-deleted lines indicates that the loss of the *var2csa* locus disrupted the organization of the underlying pattern of *var* gene minor transcripts or its stability, which could in turn disrupt or substantially alter the structured pattern of *var* gene transcriptional activation.

## The 'single' and 'many' *var* expression states are associated with differences in the overall ring-stage transcriptome

To gain additional insights into the two hypothetical *var* expression states, we employed whole transcriptome analysis of both wildtype and *var2csa*-mutated lines. Specifically, we performed RNAseq from tightly synchronized ring stage (~16 hours post-invasion [hpi]) and trophozoite stage (~34 hpi) parasite populations. We compared transcriptomes from 11 different lines, including 4 *var2csa*-mutated lines in the 'single' state, 3 *var2csa*-mutated lines in the 'many' state, and 4 wildtype lines (2 'manys' and 2 'singles'). To control for possible effects from plasmid integration, two of the wildtype lines (called WT_rif1 and WT_rif2) contain a plasmid integrated into a *rif* gene, analogous to the integration used to disrupt *var2csa* as shown in *Figure 3C*. Because we were interested in changes to expression of genes other than those that undergo clonally variant expression (*Cortés and Deitsch, 2017*), we excluded *var*, *rif*, *stevor*, and *phist* genes and performed the analysis on the remaining protein coding genes in the 3D7 genome.

As an initial assessment of gene expression in ring-stage parasites (16 hpi), we performed principal component analysis based on the mRNA levels of 5721 genes to determine if gene expression patterns are associated with the different *var* expression states. This analysis demonstrated clear clustering between populations that displayed either the 'single' or 'many' phenotype (*Figure 4E*). Further, the parasite lines with the *var2csa*-mutant 'single' or 'many' expression profiles displayed the most polarized patterns of gene expression while the wildtype lines presented intermediate expression patterns. Permutational multivariate ANOVA (ADONIS) confirmed that the *var* expression profiles contributed strongly to gene expression variability ($R^2$ = 0.42768, \*\*\*p = 0.001). This implies that the *var* gene expression state (either 'single' or 'many') is associated with changes in the overall transcriptome at the ring stage of the asexual cycle. Interestingly, this phenomenon was not observed in trophozoites (see below). Our previous qRT-PCR analysis of wildtype 'single' and 'many' parasite lines found that they display very different total *var* gene expression levels (*Figure 1C, E*). To confirm this observation, we quantified the *var* transcripts from our RNAseq analysis from all sequenced lines. Consistent with our previous results, the wildtype 'single' lines displayed higher total *var* transcripts than the 'many' lines, and these differences were magnified in the *var2csa*-mutant lines (*Figure 4F*).

## Transcriptome analysis indicates that 'single' and 'many' parasites differ in cell cycle progression

Further analysis of the transcriptional differences between the populations identified 611 differentially expressed genes (false discovery rate [FDR] <0.05) when comparing the *var2csa*-mutant 'singles' to the *var2csa*-mutant 'manys' at the ring stage (*Figure 5A*, left; *Supplementary file 1*). Specifically, we found 562 genes upregulated in 'singles' and the remaining 49 genes upregulated in 'manys', and these differences largely disappeared in trophozoites (*Figure 5A*, right; *Supplementary file 2*). Functional and gene ontology analyses of the differentially expressed genes only presented broad, relatively unspecific terms and did not identify any enriched metabolic pathways or functions (*Supplementary file 3* and *Supplementary file 4*). This result prompted us to investigate whether the observed transcriptional changes were indicative of a shift in cell cycle progression rather than specific changes in cellular functions. We therefore applied two independent methods as described by *Poran et al., 2017* and *Lemieux et al., 2009* to estimate the approximate point in the replicative cycle of each of our ring-stage parasite populations. We compared the transcriptomes obtained from our lines to hypothetical transcriptomes derived from modeled parasite populations over a simulated asexual replication time course. Both methods consistently indicated a shift in cell cycle progression, with the *var2csa*-mutant 'single' lines displaying an earlier point in the asexual cycle than the *var2csa*-mutant 'many' lines and with the wildtype lines displaying an intermediate time point (*Figure 5B, C*). This continuum in cell cycle progression was consistent for all 11 lines, despite the fact that all cultures were synchronized to 16 hpi. Once again, the *var2csa*-mutant lines occupied the extreme positions in this continuum, indicating that these lines display more polarized phenotypes (*Figure 5B, C*, *Supplementary file 5*).

To further investigate the possibility that the differences in gene expression resulted from a shift in ring-stage progression, we used the PlasmoDB database to examine the transcriptional timing of all the differentially expressed genes. We found that of the 562 genes upregulated in 'singles', 559 have higher transcription levels at 0–10 hpi compared to 10–20 hpi and similarly, all 49 genes upregulated in 'manys' have higher transcription levels at 10–20 hpi compared to 0–10 hpi (*Figure 5D, E*, *Supplementary file 3* and *Supplementary file 4*; *Bártfai et al., 2010*), providing additional evidence that the transcriptional differences of these populations likely reflect a shift in the replicative cycle. The expression levels of the differentially expressed genes for all 11 lines again confirm that the 4 wildtype lines display an intermediate level of transcription for all the differentially expressed genes (*Figure 5F*). Taken together, these analyses indicate that being in the 'single' or 'many' state correlates strongly with how quickly parasites progress through the ring stage of the cycle. In addition, the *var2csa*-mutant lines display more polarized phenotypes, suggesting that these populations are more homogenous in being either 'single' or 'many', a conclusion consistent with the hypothesis that in the absence of *var2csa*, parasites are less efficient in transitioning between states. A similar change in ring-stage progression was previously observed in response to treatment with a histone methyltransferase inhibitor (*Chan et al., 2020*) that also altered *var* gene expression patterns (*Ukaegbu et al., 2015*), as well as in parasites that display reduced sensitivity to artemisinin (*Mok et al., 2011*; *Mok et al., 2015*), indicating that how quickly parasites progress through the first half of their replicative cycle is somewhat flexible. Interestingly, these differences appear to be largely limited to progression through the ring stage. Measurement of DNA content of synchronized cultures by flowcytometry indicated that *var2csa*-mutant 'single' and 'many' populations undergo DNA replication with similar timing as wildtype parasites (*Figure 5G*), and growth assays did not detect any overall differences in the growth of the cultures (*Figure 5—figure supplement 1*), indicating that length of the complete replication cycle is not detectably different. Combined with the gene expression data obtained from RNAseq, it appears that 'single' parasites catch up to 'manys' as they begin DNA replication, thereby negating any growth advantage. These experiments indicate that the ring stages of 'single' and 'many' states are distinct beyond just *var* gene expression, and likely reflect changes in the nuclear or epigenetic organization of the parasite's genome that affect ring-stage progression and associated gene expression patterns.

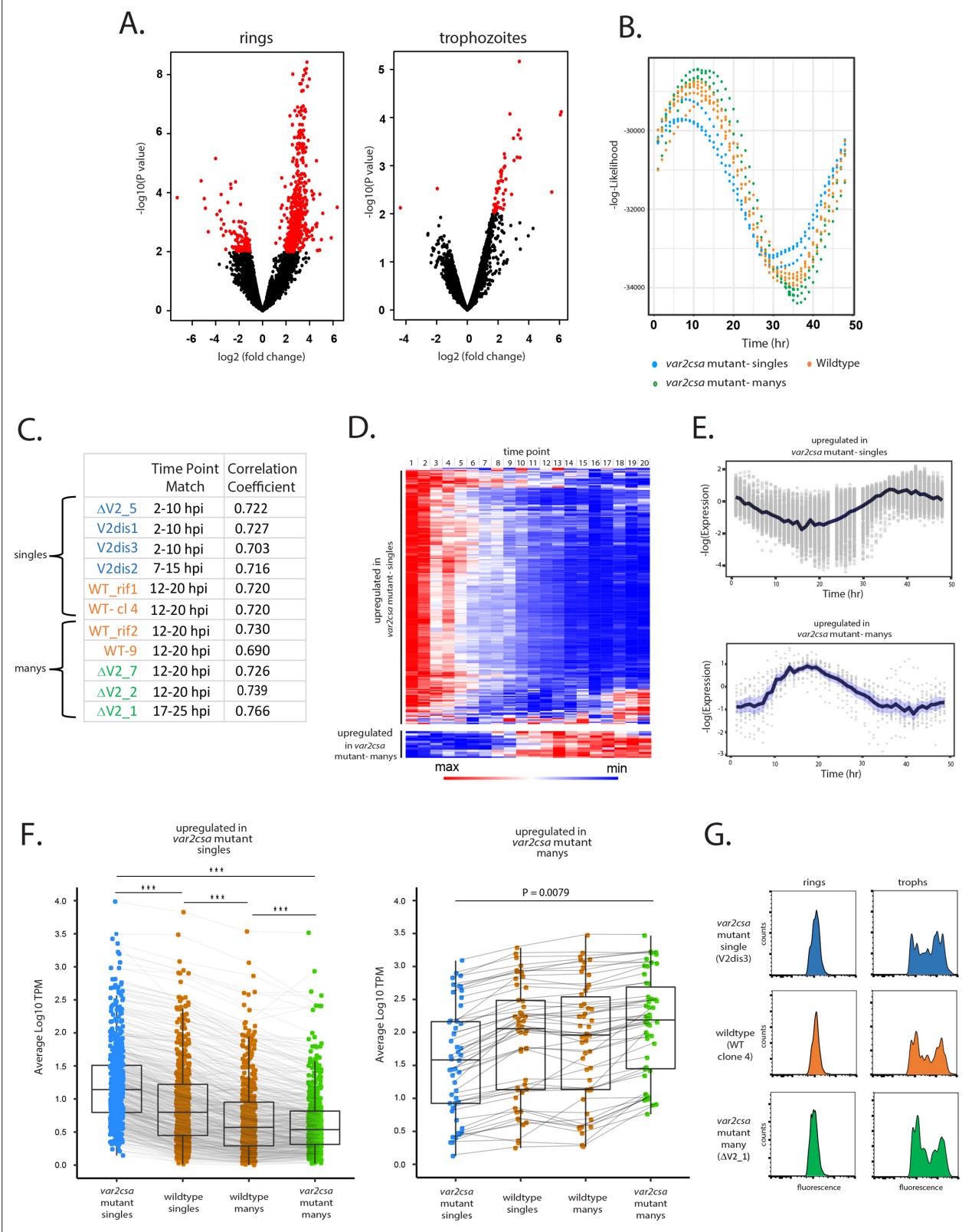

**Figure 5.** Parasites displaying 'single' vs. 'many' *var* expression profiles differ in replication cycle progression. (**A**) Volcano plots showing differentially expressed genes derived from whole transcriptome comparisons of *var2csa*-mutated 'single' and 'many' parasite lines. Differentially expressed genes are shown in red. For ring-stage parasites (left), 562 transcripts displayed higher expression levels in the single lines, while 49 were higher in the many lines. For trophozoite-stage parasites, only 51 genes were differentially expressed. (**B**) Estimation of cell cycle position using the method of

*Figure 5 continued on next page*

*Figure 5 continued*

*Lemieux et al., 2009*. Using the transcriptome profiles in *Supplementary file 8* (*Bártfai et al., 2010*), the likelihood (vertical axis) of each sample having been derived from a particular time point of the cycle (horizontal axis) is displayed. Eleven parasite lines are shown, with wildtype parasites in orange, *var2csa*-mutated, 'single' parasites in blue and *var2csa*-mutated, 'many' parasites in green type. (**C**) Table showing estimates of replication cycle progression for all 11 parasite lines. Estimates of the approximate time point within the 48-hr asexual cycle for each population were obtained by comparison with a modeled parasite population over a simulated infection time course using established datasets (*Poran et al., 2017*). The best time point match is shown for each population along with the correlation coefficient. Wildtype parasites are designated with orange type, *var2csa*-mutated, 'single' parasites with blue type and *var2csa*-mutated, 'many' parasites with green type. (**D**) Heatmap displaying changes in expression over the asexual replicative cycle for 611 differentially expressed genes according to the 48-hr asexual cycle of the *P. falciparum* HB3 transcriptome datasets defined by *Bozdech et al., 2003*; *Supplementary file 9*. Genes upregulated in *var2csa*-mutated 'single' lines are shown on top and those from *var2csa*-mutated 'many' lines are displayed on the bottom. (**E**) Average expression levels across the 48-hr asexual cycle of HB3 for 611 differentially expressed genes. Genes upregulated in *var2csa*-mutated, 'single' lines are shown on top and those upregulated in *var2csa*-mutated, 'many' lines are shown on the bottom. (**F**) Left panel, transcript expression levels for 559 differentially expressed genes that display higher expression in 'single' parasites. Average, normalized transcript levels were obtained from *var2csa*-mutated, 'single' parasites (blue), wildtype 'single' and 'many' parasites (orange, left and right, respectively), and *var2csa*-mutated, 'many' parasites (green). Box and whisker plots display the mean and standard deviation for each dataset. Right panel, analysis of transcript expression levels for 49 differentially expressed genes that display higher expression in 'many' parasites. Pairwise comparisons using *t*-tests with pooled standard deviation. p value adjustment method is from Bonferroni (significance codes: 0 '***', 0.001 '**', 0.01 '*'). (**G**) Monitoring of DNA content of infected cells by flow cytometry. Populations of *var2csa*-mutant 'singles' (top), wildtype (middle), and *var2csa*-mutant 'manys' (bottom) were tightly synchronized then assayed for DNA content by flow cytometry using Hoechst 33342 staining. Parasites were assayed at 16 and 36 hr after invasion.

The online version of this article includes the following figure supplement(s) for figure 5:

**Figure supplement 1.** Growth assays of parasites displaying the 'single' and 'many' *var* expression phenotypes.

## Discussion

The SMS model for *var* gene transcriptional switching proposes that switching expression from one *var* gene to another involves transition from transcribing a single *var* gene, to many, then back to a single gene again (*Figure 1A*). This type of two-step selection of a gene for activation is now well documented for the choice of a single transcriptionally active olfactory receptor gene in mammals (*Serizawa et al., 2004*) and more recently for expression of a single *vsg* in metacyclic African trypanosomes (*Hutchinson et al., 2021*). In both cases this involves the initial, low-level transcription of multiple genes, which is then limited to a single gene in fully differentiated cells (*Tan et al., 2015*; *Hutchinson et al., 2021*; *Hanchate et al., 2015*; *Saraiva et al., 2015*). In olfactory neurons, this coincides with the stepwise establishment of specific interchromosomal contacts and the assembly of subnuclear compartments that limit expression to a single gene (*Monahan et al., 2019*). Our results indicate that a similar process occurs in malaria parasites during *var* gene switching, a possibility that is consistent with the suggestion that individual parasites can express multiple *var* genes, a phenomenon that was previously reported using immunofluorescence and in situ hybridization (*Joergensen et al., 2010*). If correct, mutually exclusive *var* gene expression is possibly more plastic than previously assumed, although additional work at the single cell level will be required to fully characterize the extent of *var* gene activation in individual parasites in the 'many' state.

Populations of parasites in which *var2csa* was deleted or disrupted displayed slower *var* gene switching rates (*Figure 3*) and appear more polarized with respect to the 'single' vs. 'many' states (*Figure 5*), indicating that they are much less efficient in transitioning between these two states. How the *var2csa* locus contributes to transcriptional switching is not known, although given that the *var2csa* promoter region appears to be important for this function, it is tempting to propose a model of promoter competition, as has been proposed for chromosome choice during X-inactivation in mammals or *vsg* expression site choice in African trypanosomes (*Hutchinson et al., 2021*; *Constância et al., 1998*). Such models typically propose that when in the many state, multiple loci compete or 'race' for activation, with each locus expressing a low level of transcripts. When one locus reaches a threshold, transcription from the other loci is suppressed, resulting in stable monoallelic expression. If this model applies to *var* gene expression, the unique nature of the *var2csa* promoter could enable it to compete with an actively expressed *var* gene, thereby lowering its expression level to below the threshold and thus allowing the parasite to re-enter the 'many' state. This would provide an explanation for why loss of the *var2csa* locus results in stabilization of the active *var* gene and greatly reduces switching. Previous work from *Duffy et al., 2009*; *Duffy et al., 2017* showed that alterations to sequences adjacent to the *var2csa* promoter can further increase its competitiveness, making it

hypercompetitive and stably expressed, thus suppressing switching rates in a way consistent with this model. The propensities of cultured parasites to converge toward expression of *var2csa* independent of genomic rearrangement is also consistent with its highly competitive nature (*Mok et al., 2008*). To complete a switch to an alternative *var* gene, activation of *var2csa* would have to be unstable, a property that could result from its unique structure, including the presence of an uORF and an untranslated mRNA (*Amulic et al., 2009*; *Bancells and Deitsch, 2013*). While *var2csa* is unique in its structure and appears to be the dominant 'sink node' in the experiments described here, other *var* genes could also display greater propensities for activation, thereby functioning as additional nodes and providing greater structure to the network. For example, we observed frequent activation of Pf3D7_0421100 in our subcloned populations, indicating this gene could be acting as a node, at least in the context of our study.

The location of a large proportion of the *var* gene family within subtelomeric regions makes them uniquely subject to frequent recombination and deletion events, an inherent plasticity that is reflected in the variable number of *var* genes observed in the genomes of different parasites isolates (*Otto et al., 2019*). Interestingly, despite the hyper-recombinogenic nature of the subtelomeric region in which it resides, *var2csa* remains conserved in all *P. falciparum* lines examined, and this conservation extends to the related species *P. reichenowi* and *P. praefalciparum* (*Gross et al., 2021*). We hypothesize that its role in mediating expression switching selects for its preservation within the genomes of wildtypes parasites. This hypothesized additional role for *var2csa* is consistent with detection of its transcription non-pregnant individuals (*Duffy et al., 2006*; *Beeson et al., 2007*; *Rovira-Vallbona et al., 2011*) as well as activation of *var2csa* coincident with *var* gene switching during an experimental human infection (*Bachmann et al., 2019*). We previously documented frequent subtelomeric deletions that included *var* genes on other chromosomes (*Calhoun et al., 2017*; *Reed et al., 2021*), including the subtelomeric regions of chromosomes 2, 3, and 14 in the 3D7 control lines employed here (*Calhoun et al., 2017*), however none of these deletion events resulted in altered *var* switching phenotypes, indicating that the properties we describe here are unique to the *var2csa* locus. The frequently described convergence of cultured parasite populations to expression of *var2csa* could be a result of a selective growth advantage parasites acquire if the *var2csa* transcript only translates the uORF, thus saving the metabolic cost of translating a full-length PfEMP1. While feasible, if true one would expect the 'many' parasites to similarly display a selective growth advantage given their very low rate of *var* gene transcription, which we do not observe (*Figure 5—figure supplement 1*).

In addition to changing the frequency of *var* expression switching, deletion of *var2csa* also resulted in major changes in the pattern of minor *var* transcripts displayed by parasite populations (*Figure 3*). This underlying pattern of transcription has been proposed to reflect switching biases within the *var* gene network (*Recker et al., 2011*; *Horrocks et al., 2004*), thereby imposing a structure on the trajectory that *var* expression takes over the course of an infection. This type of structured genetic network could coordinate *var* gene activation throughout the parasite population without requiring communication between infected cells and thus would enable lengthy infections despite a relatively small number of variant antigen-encoding genes. The shifting pattern of *var* minor transcripts observed in wildtype parasites and shown in *Figure 2* is consistent with this hypothesis. The disruption of the pattern of minor transcripts upon deletion of *var2csa* indicates that this gene plays an important role in maintaining the structure of the *var* gene transcriptional network, although the molecular mechanism underlying this phenomenon is not known. Additional work focused on subnuclear genome organization, specifically with regard to heterochromatic regions, and the role that *var2csa* plays in this organization are likely to shed light on this process.

The switching biases observed in cultured parasites could help refine the trajectory of *var* expression over the course of an infection, as suggested by the model proposes by *Recker et al., 2011*. However, it is important to note that while *var* gene expression patterns displayed by parasite cultures likely reflect inherent switching biases, in a natural infection the host's immune system, in particular existing antibodies against PfEMP1, are likely to exert a profound selective pressure that determines which parasites can successfully establish a high parasitemia. Pre-existing anti-PfEMP1 antibodies that recognize any particular variant will select against parasites expressing its encoding *var* gene, thus strongly shaping the *var* expression pattern over the course of an infection. The importance of antibody selection and pre-existing immunity explains why expression patterns cannot be truly hardwired

and must maintain flexibility, indicating why switching biases rather than a strict switching order are ideal for maintaining chronic infections.

Given the lack of an in vivo experimental system for *P. falciparum*, it is difficult to investigate the 'single' and 'many' states in natural infections. However, an interesting case study of a semi-immune African immigrant to Europe might be informative (*Bachmann et al., 2009*). No parasites were initially detected in this patient by microscopy or by malaria antigen tests, although the individual displayed high titers of antibodies to malaria antigens indicating a significant level of immunity. Upon splenectomy, parasitemia became evident and increased to high levels, indicating a latent infection existed prior to removal of the spleen. The parasites observed in the peripheral circulation represented all stages of the asexual cycle, indicating a lack of cytoadherence, and in vitro binding assays failed to detect adhesion to various endothelial receptors. Further, molecular analysis indicated that these parasites did not express *var* genes at a detectable level. However, upon transfer to in vitro culture, *var* gene expression became easily detectable and the infected cells became cytoadherent. This is reminiscent of our observations that parasites can transition between high-level expression of individual *var* genes (the 'single' state) and very low-level *var* gene expression (the 'many' state). A similar example of parasites lacking expression of surface antigens upon splenectomy was observed in monkeys infected with *Plasmodium knowlesi* (*Barnwell et al., 1983*), suggesting that this might be a more general phenomenon among malaria parasite species that display cytoadherence. It is tempting to speculate that in the presence of high titers of anti-PfEMP1 antibodies, parasites in the 'many' state are selected for and thus represent the dominant population found in semi-immune, asymptomatic infected individuals. Consistent with this hypothesis, Kho et al. more recently reported that the vast majority of the parasite biomass in asymptomatic infections is found in the spleen (*Kho et al., 2021*), indicative of non-cytoadherent parasites and potentially reflecting lower expression of PfEMP1. Similarly, a recent study of parasites obtained from asymptomatic patients at the end of a dry season detected a lower level of *var* expression than observed in parasites from symptomatic patients, although this difference was not statistically significant (*Andrade et al., 2020*). Thus, it is possible that in individuals with significant levels of immunity, parasites in the 'many' state can persist at a low level, escaping anti-PfEMP1 antibodies through repressed *var* gene expression and maintaining very low parasitemias.

# Materials and methods

## Key resources table

| Reagent type (species) or resource | Designation | Source or reference | Identifiers | Additional information |
|---|---|---|---|---|
| Gene (*Plasmodium falciparum*) | *var2csa* | EuPathDB | PF3D7_1200600 | |
| Strain, strain background (*P. falciparum*) | NF54 | *Delemarre and van der Kaay, 1979* (PMID:390409) | NF54 | |
| Strain, strain background (*P. falciparum*) | 3D7 | *Walliker et al., 1987* (PMID:3299700) | 3D7 | |
| Transfected construct (*P. falciparum*) | pL6_eGFP | Ghorbal et al. (PMID:24880488) | CRISPR-targeting plasmid | |
| Transfected construct (*P. falciparum*) | pUF1_Cas9 | Ghorbal et al. (PMID:24880488) | CRISPR-targeting plasmid | |
| Software, algorithm | aligner HISAT2 | *Kim et al., 2019* (PMID:31375807), *Afgan et al., 2018* (PMID:29790989) | v.2.2.0 | |
| Software, algorithm | featureCounts | *Liao et al., 2014* (PMID:24227677) | Package Rsubread v.2.0.1 | |
| Software, algorithm | DESeq2 | *Love et al., 2014* (PMID:25516281) | v.3.14 | |
| Software, algorithm | EdgeR | *Chen et al., 2016* (PMID:27508061), *McCarthy et al., 2012* (PMID:22287627), *Robinson et al., 2010* (PMID:19910308) | v.3.14 | |
| Software, algorithm | R | Rstudio (1.4.1717) | v.4.1.0 | |

*Continued on next page*

*Continued*

| Reagent type (species) or resource | Designation | Source or reference | Identifiers | Additional information |
|---|---|---|---|---|
| Software, algorithm | vegan | Community Ecology Package | v.2.5-7 | |
| Software, algorithm | PERMANOVA | Community Ecology Package | v.2.5-7 | |
| Software, algorithm | heatmap.2 | gplots | v.2.3.0 | |

## Parasite culture

All parasite strains were derived from the reference strain NF54/3D7 and maintained in RPMI 1640 supplemented with 0.5% AlbuMAX II (Invitrogen) in an atmosphere of 5% $O_2$, 5% $CO_2$, and 90% $N_2$ at 37°C and 3–5% hematocrit. Strain identify was confirmed by whole-genome sequencing and any contaminating mycoplasma were removed at the beginning of the study as described (*Singh et al., 2008*). Transgenic lines were maintained under continuous drug selection. Clonal parasite lines were obtained by limiting dilution *Kirkman et al., 1996* in 96-well plates. For analysis of gene expression in subcloned populations, parasites were grown for 5 weeks from the initial isolation of individual parasites. Daily parasitemias were monitored using a Cytek DxP12 flow cytometer.

## Generation of transgenic lines

Parasites with chromosomal truncations that included the *var2csa* locus were generated and described in a previous study (*Zhang et al., 2019*). Specific disruption of the *var2csa* locus was performed via CRISPR-targeted plasmid integration. Briefly, parasites were transfected by electroporation (*Wu et al., 1996*; *Deitsch et al., 2001*) using derivatives of the plasmids pL6_eGFP and pUF1_Cas9 for CRISPR/Cas9-based genome editing as described by *Ghorbal et al., 2014*. Homology blocks were PCR amplified from 3D7 genomic DNA, using specific primers flanked by 15-bp overlapping regions from pL6-*var2csa*-promoter-deletion plasmid and pUF1_Cas9 (list of primers, see *Supplementary file 6*) that allowed the cloning by infusion cloning (Clontech, Takara Bio USA, Mountain View, CA, USA). Plasmid integration was validated by PCR across the site of integration and whole-genome sequencing.

## RNA extraction, cDNA synthesis, and quantitative RT-PCR

Parasite RNA was extracted from synchronized late ring-stage parasites as described previously (*Dzikowski et al., 2006*). Briefly, ring-stage parasites were collected using a double synchronization approach. Cultures were initially synchronized using 5% sorbitol to select for ring stages. Thirty-six hours later, late-stage infected erythrocytes (42–48 hpi) were collected using percoll/sorbitol gradient centrifugation and allowed to reinvade overnight. Ring-stage parasites were collected for RNA extraction 18 hr after enrichment for late-stage infected erythrocytes (12–18 hpi). After an additional 20 hr in culture, trophozoites were collected for RNA isolation. Synchronization was verified by microscopy and flow cytometry. RNA was extracted with TRiZol (Invitrogen) and purified on PureLink (Invitrogen) columns following the manufacturer's protocols. Isolated RNA was treated with Deoxyribonuclease I (DNase I) (Invitrogen) to degrade contaminating genomic DNA. cDNA was synthesized from approximately 800 ng of RNA in a reaction that included Superscript II RNase H reverse transcriptase (Invitrogen) as described by the manufacturer. Control reactions in the absence of reverse transcriptase were employed to verify a lack of gDNA contamination. We employed the qRT-PCR *var* primer set of Salanti et al. to detect transcript levels from all *var* genes (*Salanti et al., 2003*). This primer set was specifically designed and tested to enable absolute quantification of transcript levels. Additional primers were designed and applied to increase the accuracy (list of primers, see *Supplementary file 6*). Quantitative PCR was performed using a Quant Studio 6 Flex 489 realtime PCR machine (Thermo Fisher) using iTaq Sybr Green (Bio-Rad). Quantities were normalized to seryl-tRNA synthetase (PF3D7_0717700). ΔCT for each individual primer pair was determined by subtracting the individual CT value from the CT value of the control and converting to relative copy numbers with the formula $2^{\Delta CT}$. All qRT-PCR assays were performed in a 384-well format PCR machine enabling duplicate or triplicate runs performed simultaneously. Biological replicates were prepared from independent RNA extractions. Relative copy numbers for each *var* gene were determined in Microsoft Excel

and transcriptional profiles of individual genes are presented as pie charts or as bar graphs. In bar graphs, standard deviations from biological replicates are shown with error bars.

## RNA sequencing and analysis

Parasite-infected RBCs from highly synchronous cultures containing ring or trophozoites stage parasites were collected for RNAseq. Following RNA isolation, RNA concentrations were measured using the NanoDrop system (Thermo Fisher Scientific, Inc, Waltham, MA). Total RNA integrity was checked using a 2100 Bioanalyzer (Agilent Technologies, Santa Clara, CA). rRNA removal, preparation of an RNA sample library and final cDNA library were completed and the libraries pooled for sequenced using an Illumina HiSeq4000 sequencer with paired-end protocol performed by the Genomics Core Laboratory at Weill Cornell Medicine. The raw reads that passed quality control were mapped to the *P. falciparum* genome (PlasmoDB-9.0_Pfalciparum3D7_Genome) by aligner HISAT2 (v.2.2.0) (*Kim et al., 2019*; *Afgan et al., 2018*). Based on the alignment, featureCounts (Package Rsubread v.2.0.1) (*Liao et al., 2014*) was used to generate a raw counts matrix for differential expression analysis. We independently performed both DESeq2 (v.3.14) (*Love et al., 2014*) and EdgeR (v.3.14) (*Chen et al., 2016*; *McCarthy et al., 2012*; *Robinson et al., 2010*) for the differential expression analysis. Genes with a false discovery rate of ≤0.05 with a mean fragments per killobase of transcript per million mapped fragments (FPKM) >5 in at least one strain were called significant. To exclude the obvious impact of antigenic variant genes, we excluded *var*, *rif*, *stevor*, and *phist* genes from the differential expression analysis. The analysis was conducted with R (v.4.1.0) in Rstudio (1.4.1717). RNA sequencing was performed by the Weill Cornell Genomic core with the following parameters. All samples must have an RIN value greater than 8 for library synthesis. Ribosomal RNA was removed from total RNA using the Ribo Zero Gold for human/mouse/rat kit (Illumina). Using the TruSeq RNA Sample Library Preparation v.2 kit (Illumina), RNA was fragmented into small pieces using divalent cations under elevated temperature. Cleaved RNA fragments were copied into first-strand cDNA using reverse transcriptase and random primers. Second-strand cDNA synthesis followed, using DNA Polymerase I and RNaseH. The cDNA fragments then went through an end repair process, the addition of a single 'A' base and ligation of adaptors. The products were then purified and enriched with PCR to create the final cDNA library. Libraries were pooled and sequenced on an Illumina HiSeq4000 sequencer with 100-bp Paired End Sequencing (PE100). Total reads and mapped reads for each sample are shown in *Supplementary file 7*.

## Statistical analysis

Statistical analysis was performed in R (v.4.1.0). We assessed beta diversity of our samples' *var genes* expression profiles using NMDS ordination. NMDS plots and beta dispersion were generated in vegan: Community Ecology Package (v.2.5-7) using Bray–Curtis dissimilarity. Permutational multivariate analysis of variance (PERMANOVA) was performed using the 'adonis' function in vegan. Heatmaps were generated by heatmap.2 from gplots (v. 2.3.0).

## Flow cytometry and assays for cell cycle progression

Progression through the cell cycle was determined by flow cytometric analysis of parasite DNA content as previously described (*Grimberg, 2011*). Briefly, parasites were tightly synchronized were stained at 37°C with 16 µM Hoechst 33342 for 30 min at 1% hematocrit in incomplete media followed by a single wash in phosphate-buffered saline (PBS). Cells were then diluted to 0.1% hematocrit in PBS and analyzed using a Cytek DxP11 flow cytometer for Hoechst 33342 DNA staining (375 nm laser, 450/50 emission filter).

## Growth assays

Cultures were adjusted to 0.05–0.5% parasitemia, 5% hematocrit in a total culture volume of 5 ml. Parasitemia was obtained daily by flow cytometry and verified by thin smear stained with Giemsa (Sigma). Parasitemia was allowed to increase over time until reaching a level greater than 0.5–1%, at which time cultures were diluted 1 in 10 in culture media containing uninfected RBCs. Changes in parasitemia were calculated by multiplying the daily parasitemia by the exponential dilution factor and the data were graphed on a log scale over time.

## Materials availability statement

Data deposition: Whole-genome sequence and transcriptome data are available at the BioProject database of the NCBI. The genome sequencing data can be accessed at this link: http://www.ncbi.nlm.nih.gov/bioproject/515738. The RNAseq data can be accessed at this link: https://www.ncbi.nlm.nih.gov/bioproject/?term=PRJNA802886. All genetically modified parasite lines are available from the authors upon request.

## Acknowledgements

The Department of Microbiology and Immunology at Weill Medical College of Cornell University acknowledges the support of the William Randolph Hearst Foundation. This work was supported by grants AI 52390 and AI99327 from the National Institutes of Health to KWD. JEV was supported by training grant T32GM008539 from the National Institutes of Health to Weill Cornell Graduate School of Biomedical Sciences and by fellowship number F31AI164897 from the National Institutes of Health. FF received support from Early Postdoc Mobility grant P2BEP3_191777 from the Swiss National Science Foundation. KWD is a Stavros S Niarchos Scholar and a recipient of a William Randolf Hearst Endowed Faculty Fellowship. The funders had no role in the study design, data collection and analysis, decision to publish, or preparation of the manuscript.

## Additional information

### Funding

| Funder | Grant reference number | Author |
| --- | --- | --- |
| National Institute of Allergy and Infectious Diseases | AI 52390 | Kirk W Deitsch |
| National Institute of Allergy and Infectious Diseases | AI99327 | Kirk W Deitsch |
| National Institutes of Health | T32GM008539 | Joseph E Visone |
| Swiss National Science Foundation | P2BEP3_191777 | Francesca Florini |
| National Institutes of Health | F31AI164897 | Joseph E Visone |

The funders had no role in study design, data collection, and interpretation, or the decision to submit the work for publication.

### Author contributions

Xu Zhang, Conceptualization, Data curation, Formal analysis, Investigation, Methodology, Writing – original draft, Writing – review and editing; Francesca Florini, Data curation, Formal analysis, Funding acquisition, Investigation, Methodology, Writing – review and editing; Joseph E Visone, Data curation, Formal analysis, Investigation, Methodology, Writing – review and editing; Irina Lionardi, Formal analysis, Writing – review and editing; Mackensie R Gross, Data curation, Methodology; Valay Patel, Data curation; Kirk W Deitsch, Conceptualization, Data curation, Formal analysis, Supervision, Funding acquisition, Validation, Investigation, Methodology, Writing – original draft, Project administration, Writing – review and editing

### Author ORCIDs

Francesca Florini ⓘ http://orcid.org/0000-0003-2579-3820
Kirk W Deitsch ⓘ http://orcid.org/0000-0002-9183-2480

### Decision letter and Author response

Decision letter https://doi.org/10.7554/eLife.83840.sa1
Author response https://doi.org/10.7554/eLife.83840.sa2

## Additional files

### Supplementary files

• Supplementary file 1. List of differentially expressed genes when comparing *var2csa*-mutant 'single' lines to *var2csa*-mutant 'many' lines at ring stages. Genes upregulated in 'singles' are denoted in blue while those upregulated in 'manys' are denoted in green. log2 FC = logFold Change, logCPM = log(normalized mean Count per Million), LR = likehood ratio, FDR = adjust p value.

• Supplementary file 2. List of differentially expressed genes when comparing *var2csa*-mutant 'single' lines to *var2csa*-mutant 'many' lines in trophozoite stage. Genes upregulated in 'singles' are denoted in blue while those upregulated in 'manys' are denoted in green. logCPM = log10(normalized mean Count per Million), LR = likelihood ratio, FDR = adjust p value.

• Supplementary file 3. Annotation numbers and gene ontology terms for each of 562 genes upregulated in *var2csa*-mutant 'single' lines when compared to *var2csa*-mutant 'many' lines. Also shown are expression values described for each gene across the 48-hr asexual replication cycle as established in the published dataset from Bartfai et al. These values were used to establish the approximate position within the standard asexual cycle for each sample.

• Supplementary file 4. Annotation numbers and gene ontology terms for each of 49 genes upregulated in *var2csa*-mutant 'many' lines when compared to *var2csa*-mutant 'single' lines. Also shown are expression values described for each gene across the 48-hr asexual replication cycle as established in the published dataset from Bartfai et al. These values were used to establish the approximate position within the standard asexual cycle for each sample.

• Supplementary file 5. Detailed likelihood of each sample time point using the method described by Poran et al. All time points refer to the 48-hr asexual replication cycle.

• Supplementary file 6. Primers used during the study. The top four primers were used for PCR amplification of specific homology blocks during the construction of the pL6-*var2csa*-promoter deletion plasmid. (A) Primers for plasmid pL6-*var2csa*-promoter-deletion and pLcas9 construction. (B) Additional quantitative real-time RT-PCR (qRT-PCR) primers for *var* transcription detection.

• Supplementary file 7. Summary of the total number of reads and mapped reads for the transcriptome analysis of various 'single' and 'many' expression lines, including var2csa-mutant lines (ΔV2_1, V2dis3, ΔV2_7, V2dis1, V2dis2, ΔV2_5, and ΔV2_3) and wildtype lines (Subclone 3.1, WT_7, WT_rif2, and WT_rif1). Analysis of ring stages is shown on the left and trophozoite stages are on the right.

• Supplementary file 8. RNAseq dataset from *Bártfai et al., 2010* used as the reference for estimating the replication cycle time point of each sample population using the method of *Poran et al., 2017*. Transcription levels (FPKM) are shown for each gene through eight time points of the *Plasmodium falciparum* (3D7) asexual cycle.

• Supplementary file 9. Microarray dataset of *Plasmodium falciparum* (HB3) gene expression from *Bozdech et al., 2003* used as the reference to estimate the replication cycle time point of RNAseq samples using the method of *Lemieux et al., 2009*. The form was kindly provided by Dr. Silvia Portugal.

• MDAR checklist

### Data availability

Whole-genome sequence and transcriptome data are available at the BioProject database of the NCBI. The genome sequencing data can be accessed at this link: http://www.ncbi.nlm.nih.gov/bioproject/515738. The RNAseq data can be accessed at this link: https://www.ncbi.nlm.nih.gov/bioproject/?term=PRJNA802886.

The following dataset was generated:

| Author(s) | Year | Dataset title | Dataset URL | Database and Identifier |
|-----------|------|---------------|-------------|-------------------------|
| Zhang X, Deitsch KW | 2022 | A coordinated transcriptional switching network mediates antigenic variation of human malaria parasites | https://www.ncbi.nlm.nih.gov/bioproject/?term=PRJNA802886 | NCBI BioProject, PRJNA802886 |

The following previously published dataset was used:

| Author(s) | Year | Dataset title | Dataset URL | Database and Identifier |
|---|---|---|---|---|
| Zhang X, Deitsch KW | 2019 | Rapid antigen diversification through mitotic recombination in the human malaria parasite *Plasmodium falciparum* | https://www.ncbi.nlm.nih.gov/bioproject/PRJNA515738 | NCBI BioProject, PRJNA515738 |

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
