## [Editor Report]

This is an important study addressing the mechanisms of variant gene expression and switching in the malaria parasite *Plasmodium falciparum*. The work provides solid evidence supporting the existence of a non-random, highly structured switch pathway for var genes and identifies one var gene, called var2csa, as a sink node in this network. The findings consolidate previous observations and further the understanding of the enigmatic mechanism that underlies the regulation of the var gene family and antigenic variation in *P. falciparum*, which is paramount for immune evasion and acquired immunity and further influences malaria pathology.

---

## [Decision Letter]

[Editors' note: this paper was reviewed by Review Commons.]

Thank you for submitting your article "A coordinated transcriptional switching network mediates antigenic variation of human malaria parasites" for consideration by *eLife*. Your article has been reviewed by two reviewers (including one from the initial evaluation at Review Commons), and the evaluation at *eLife* has been overseen by a Reviewing Editor and Dominique Soldati-Favre as the Senior Editor.

Based on the previous reviews and the revisions, the manuscript has been improved but there are some remaining issues that need to be addressed, as outlined below.

Recommendations for authors

1. For many clonal lineages, only pie charts are provided for visual inspection and categorization into 'single' or 'many'. However, this also appears to be largely dependent on overall var gene expression, as in Figure S1A, subclone NF54-5 has dominant expression of a single variant and low overall var gene expression and was therefore classified as 'many'. The authors should provide expression levels as displayed in Figure 1C for all subclones analyzed in the study. These graphs could be displayed next to the corresponding pie chart, or as supplemental figures.

Importantly, an unbiased approach to categorizing parasites into different states would be desirable. Perhaps the combination of a diversity index and total var gene expression could provide sufficient discrimination.

2. The transcriptomics data reveal that "manys" progress through the first half of the replicative cycle faster, yet the "singles" catch up in the second half, and thus both populations have equivalent replication cycle lengths. The authors should discuss further why 'many' parasites have no growth advantage over several parasite generations, despite faster ring stage progression. Do the 'single' parasites catch up with this lag later? Giemsa smears, in addition to growth curves, would be useful to better document progression through the IDC of 'single' versus 'many' parasites.

3. The authors should discuss how the switch model proposed by Recker et al. could work with only a single dominant sink node that is barely inactivated during the cultivation of the parasites in vitro. In fact, what about the PF3D7_0421100 gene, which is also frequently activated and stably expressed in many subclones in different generations (Figure 2B)? Could this also be a sink node?

4. Reviewers appreciated that the authors have made an effort to improve the readability and presentation of the data. However, they also noted that there is still room for improvement.

a) The labeling of the heatmaps and bar graphs is not consistent with respect to the order of var genes and the var groups are labeled twice in the heatmaps (largely on the left side and after each gene).

b) There are some inconsistencies: cultivation days for the same clonal lines are not identical in Figures 2A and 3A; clonal line V2dis2 is classified as "many" in Figure 4A but as "single" in Figures 4E and F.

c) Why do the authors not show var2csa expression in the wild-type heat maps, but in the pie charts?

d) The phylogenetic trees of clones and subclones are partly redundant, but an overview including all clonal lineages, e.g. also clones 1, 2, 4, and 5, is still missing (could be included as a Supplemental figure).

e) Var1 should not be labeled as group D, which has been shown not to exist. The authors also do not explicitly designate var3 type genes, and the reviewers suggest including var1 and var3 in group A and more accurately labeling them var1 and var3 variants.

f) The similarity of the minor transcripts between Figure 2A and C is difficult to judge from the heat maps. Why did the authors not include the subclones from Figure 2C in the Bray-Curtis dissimilarity analysis shown in Figures 4C and D?

5. After disruption of the var2csa promoter, do the authors still see var2csa expression, or can they confirm its absence?

---

## [Author Response]

Reviewer #1 (Evidence, reproducibility and clarity (Required)):Summary:Xu and colleagues studied to unravel the program that underlies the antigenic switching mechanism between members of the var gene family, which encode major virulence factors that help evade immune clearance during natural infection. The authors propose that a unique member of the family, var2csa, is central to a hierarchical switching program, which disruption of the loci results in the impairment of the program.Major comments:A) On data presentationOne of the confusions, if not annoyance, for any reader to navigate through the figures and data is the lack of standardization. The presented figures give an impression of poor stringency during the construction especially regarding the finer details.For example, the annotation of the different clones used in the study appear to be unmethodical. Ideally, the annotation of the clone should reflect information of the parental genetic background and the history of the cloning procedures. For example in Figure 1 the clones are annotated as C3C6…whereas in Figure 2 there are clones being annotated as A3.B8. Both tend to suggest the parasite have been cloned twice but named in a different way. Furthermore, A3.B8 and A3.C10 intuitively suggest both being a subclone of A3, yet they are stated not to be from the same parent population (line 221). Also on Figure 2, A3 and 3A are just too similar to give rise to confusion. It is advised to re-name all the clones methodically.

We now appreciate that the original naming scheme applied to the clones was confusing and difficult to follow. We have therefore renamed all the clones described in the manuscript and used names that reflect their origins. We hope this clarifies the logic and progression of the paper.

It is rather inappropriate and potentially biased to calculate column z-score using the relative copy numbers of different var genes in the heatmaps. Since Cq values generated from sybr green signal are dependent on the amplicon size, even different primers on the same var gene can give different relative copy numbers. Relative copy numbers can be used to calculate row z-score, but column z-score requires the use of absolute quantitation method. Furthermore, the purpose of the heatmaps used in this study is to illustrate the abundance of different minor transcripts. Since the authors do not intend to demonstrate the relationship between the clones nor the different var genes, ordering the genes by hierarchical clustering is not useful. It is more preferable to order the genes in a standardized manner (for example by the ups type), so that they can be compared between figures. The omitted dominant var genes can also be denoted by a special color in the heatmap. This can maximize data retention and the readability of the heatmaps.For example, on line 183-185, "the entire var gene family is not equally represented […] biased towards certain var genes." It is difficult to appreciate this statement because it is impossible to assess such bias using the current heatmaps.

We appreciate the concerns of the reviewer and have modified the figures appropriately. As suggested by the reviewer, the heatmaps in Figures 2 and 4 are no longer arranged using hierarchical clustering and instead have been restructured according to ups type. This provides a standardized structure and enables easy comparison of all the figures, as the reviewer had hoped. Patterns are now discernable. We thank the reviewer for this suggestion. The reviewer was also concerned about the calculation of column z-scores derived from relative copy numbers in the construction of the heatmaps, particularly if the amplicons from each var gene are not of equal length. However, the primer set designed by Salanti et al. (Mol Micro, 2003) was specifically designed and tested to enable absolute quantification. Because we intend to compare patterns across the different clones, we prefer this method for creating the heatmaps. We now specifically mention this in the Results section and the Methods section (see lines 131-134 and 601-608).

The timepoints and clones chosen for var profiling appear to be arbitrary. Different timepoints (Figure 3A and 4A) and random removal/ addition of clones (among Figure 4A,4B and 4F) are seen without proper justification.

The clones chosen for the var profiling were not arbitrary, although with the previous naming scheme this might have been difficult to follow. We hope the new naming organization will make our logic clear. The timepoints for analysis are constant in Figure 2A. The reviewer is correct that timepoints shown in Figures 3 and 4 are not constant. While perhaps not ideal, this reflects the evolution of the experiment over the years in which the project was conducted and does not alter the conclusions of the experiments. The time points are now clearly marked in each figure.

Experimental details regarding technical/ biological repeats are lacking. For example, it will be important to state how many times the CRIPSR protocol was repeated?

The var2csa deletion (via chromosome truncation) resulted from multiple independent events that were previously described in detail in Zhang et al. PLOS Bio, 2019. Clones derived from three of these independent events were fully characterized in the previous paper and were used in the current manuscript. All displayed the same phenotype. The CRISPR-mediated integration was repeated in both 3D7 and NF54, and all clones provided the same phenotype. We have added to the text to make this clearer (see lines 260-271).

B) On experimental designIn my opinion, the conclusion of the manuscript can be substantially strengthened if a control parasite, which has the promoter region of a random subtelomeric var gene deleted, was also generated and analysed. As is now, the statement on line 319-321, "These data indicate […] no other var gene within the 3D7 genome can easily substitute upon var2csa deletion.", are not necessarily supported. As one can easily argue that the same switch and convergent impairment maybe readily observed if any var gene is deleted. Indeed, ref14 cited in this manuscript also reported a complete switch impairment concommitant to the subtelomeric deletion of chr 2 and 9, dismissing the reduced-switching phenotype as a specific effect upon var2csa deletion as stated on line 321-323. It is understandable that it can be challenging to generate and analyse this control parasite due to the long time-frame of the experiment. The authors should at least address this in the discussion.

The reviewer makes a good point here that we did not discuss in the original version of the manuscript. In previously published work, we had deleted several var genes in alternative positions in the genome and did not observe the switching impairment described here. In addition, the 3D7 reference line that is frequently used throughout the world has subtelomeric deletions of var genes on both ends of chromosome 14 (Calhoun et al., 2017), but appears to undergo var transcriptional switching. We now discuss this and provide citations in the Discussion section (see lines 478-482). We also added two additional control lines to the RNAseq analysis (Figure 5) in which we disrupted a rif gene near var2csa. These lines clearly display a wildtype phenotype as expected. We also now note in the discussion that reference 31 (Mok et al) concluded that switching to var2csa was independent of genomic rearrangement (lines 461-463) and mention the work of Duffy et al. (lines 458-461), which similarly reported a genomic rearrangement adjacent to var2csa that resulted in var2csa activation leading them to conclude that the effect resulted from the cis-sequence surrounding var2csa, a hypothesis consistent with our model indicating that var2csa is highly competitive. We think these observations are entirely consistent with one another and together lead to a more refined model of how the var network is coregulated.

C) On data interpretationStochastic switching and subsequent clonal selection is a vying hypothesis of var switching program regime. While the conclusion made by the authors that var2csa is central to a switching hierarchy is fair and reasonable. It is also possible that the "sink" property of var2csa is due to in vitro selection. Var2csa expressing parasites may gain a growth advantage because as the authors pointed out, it contains a unique uORF that prevents the unnecessary translation of VAR2CSA in vitro and diverts the much needed resources for growth especially during early stage where heamoglobin digestion is limited and nutrient import machinery still nascent. Deletion of var2csa appears to cause inefficient switching may simply because no parasite, regardless of the var profile, gain any growth advantage to be selected.Results from Figure 5 that show "many"s, which express lower level of total var, are always progressing faster during early stage support such a case.One experiment that can further validates the authors claim is to generate a parasite with point mutation on the start codon of the var2csa uORF. Again such manipulation will be challenging and time-consuming, therefore, the authors should at least discuss this since stochastic switching and selection is indeed a more prevailing view among the community.

The reviewer describes a very plausible model suggesting that energy conservation from not expressing PfEMP1 could lead to convergence to var2csa. The more rapid progression through the first half of the replicative cycle by “manys” is consistent with this hypothesis. To address this possibility, we generated growth assays for “single” and “manys” as well as parasites in which var2csa had been deleted and detected no difference in growth rates (see new Supplementary Figure 1). We also observed that while “manys” progress through the first half of the replicative cycle faster, the “singles” catch up in the second half, and thus both populations have equivalent replication cycle lengths. We have added a flow cytometry analysis of DNA replication to Figure 5 to show that all clones replicate their DNA in a similar time frame suggesting that the more rapid progression through the early stages of the asexual cycle does not slow replication. We now discuss these ideas in the Discussion section (see lines 482-488).

The reviewer’s suggestion regarding creating a point mutation in the start codon of the var2csa uORF is an excellent idea and something we had previously done for a different line of investigation. This generates a profound phenotype unrelated to what we are describing in this paper and thus would be uninformative for the current manuscript. This will be described in an independent submission.

Minor comments:Specifically:1. In Figure 1C, it is not known what the values of the y-axis represent.

This has been corrected.

2. The endogenous control for the qPCR has not been specified (indeed not even in the method section).

qRT-PCR reactions were normalized to seryl-tRNA synthetase (PF3D7_0717700). This is now stated in the Methods section (see line 610).

3. For Figure 2C, importantly, none of the gene ID corresponds to a var gene. It is recommended to double-check all the source data used to generate all the figures.

As described above, all the heatmaps have been restructured, and the gene annotation numbers are now correct and organized according to ups type.

4. For Figure 3B, it is unclear the purpose for this panel. The authors have not seemingly profiled the var transcriptome of this parasite. So it is irrelevant for this manuscript.

We apologize for the confusion. Figure 3B and C display two alternative methods that both disrupted the function of var2csa as a regulator of var gene switching. Indeed, the transcriptomes of parasites derived using both of these methods are analyzed in Figures 4 and 5. We have improved the description in the text to make this clear (see lines 260-271).

5. In Figure 4A, the parasites studied are clones derived from a var2csa knockout population, whereas in Figure 4B, the parasites are claimed to be subclones derived from a clonal parasite (line 355-357). However, since some of the annotations overlap between the figs, it appears that Figure 4B is only an expansion of Figure 4A using the same parasites (at least B10, F11 and G91). However, the pie charts in Figure 4B are utterly confusing, the pie chart for G91 corresponds to the one after 1 year in Figure 4A, whereas for F11 it corresponds to the initial time point, but it resembles none of the timepoints for B10. Overall, better clarification is needed.

We agree that the original notation in the figure was unclear and confusing. As mentioned above, we have restructured the naming scheme for all the clones described in the manuscript. In addition, Figure 4A now separates the analysis for parasites derived from the different methods used to disrupt var2csa.

6. On line 359, please describe what pattern was observed.

We now arrange the gene expression profiles according to ups type rather than hierarchical clustering, therefore this section has been rewritten. We now emphasize that upon disruption of var2csa, the pattern of minor transcripts is more divergent and dispersed over time or when comparing closely related subclones, as quantified in Figure 4C.

7. For Figure 5A, I believe the x-axis should be log2FC rather than logFC, also, the p-value cutoffs were differentially applied between rings and trophozoites stage. Accordingly, the differentially regulated genes in trophozoites stage should be more than 3.

The reviewer is correct and we have corrected the figure as suggested.

8. For fig5D, the time point in the left panel should be labeled as h.p.i rather than being categorical in order to be comparable with the right panels. The panels on the right should have the same range on the y-axis.

We agree with the reviewer that it would be best to have the two figures using comparable time points, however the figures were derived from different previously published datasets, and thus we are unable to convert the data. We have now separated the analysis into two separate panels to avoid confusion.

9. For fig5E, it is much more appropriate to use TPM than CPM (as in fig4F). Since different genes have different length, CPM does not normalize for such difference.

We agree with the reviewer and have corrected the figure as suggested.

10. Fig5F intends to show single-cells that express multiple var genes have lower overall var transcription than cells with dominant var expression. It is therefore more meaningful to compute total var genes abundance on the y-axis. Furthermore, it is unclear what the value actually represents on the y-axis.

At the suggestion of one of the other reviewers, the analysis of the previously published single cell RNAseq data has been removed.

Reviewer #1 (Significance (Required)):The conception of this study is based on previous observation and the study employed methods that are commonly used in studies of similar line of research. The conclusion made by the authors is fair.Findings of this study consolidate many previous observations and further the understanding of the enigmatic mechanism that underlies the regulation of var gene family.Audience:It will be of interest for community members with lines of research focusing on surface antigens and gene regulation in the parasite. Furthermore, interest can potentially broaden to the community members within malaria research, since var genes encode major virulence factors.Field of expertise:Malaria molecular biology, Gene regulation, PfEMP1 regulation.Reviewer #2 (Evidence, reproducibility and clarity (Required)):In this study, Zhang and colleagues investigate the pattern of antigenic switching in the human malaria parasite Plasmodium falciparum. Antigenic switching between members of the var gene family as the main mechanism of immune evasion allows the parasite to maintain long infectious periods. However, the relatively limited repertoire of var genes requires some form of organisation to prevent premature exposure to the immune system. In a previous study based on in vitro data and mathematical modelling, Recker et al. hypothesised a non-random switching network involving an expansion and contraction of var transcript diversity, with some variants acting as a source and others as a sink for diversity. Using long-term cultured *P. falciparum* parasite lines, Zhang and colleague provide experimental evidence for the hypothesised switching network and further elucidate var2csa as playing a central role in var gene switching.Overall, this is a well written manuscript with results presented in a very clear fashion and following a logical structure. The amount of work that must have gone into this is also evident and admirable. My main comments are relatively minor:- l74: it would be nice to provide some reference for repeated waves of (in vivo) parasitaemias being dominated by single or just a few variants

We agree and have added appropriate references to this paragraph (see line 75 and references 19-21).

- l74-76: could mention that previous modelling studies have shown that structured switching is not strictly necessary (although appear to be optimal in semi-immune hosts)

We have modified this description as suggested by the reviewer (see lines 75-78).

- l77: there seems to be some evidence for cell-to-cell communication in *P. falciparum* (Regev-Rudzki et al., 2013); although whether this could influence var gene transcription / switching is not clear.

We agree and have added appropriate references to this paragraph (see lines 78-80 and references 25 and 26).

- l78 / general: what I was missing was more of a discussion of how results presented here align with observed in vivo switch hierarchies; that is, genetic switching might just be one side of the coin, with immune selection possibly playing another important role

We agree with the reviewer that any discussion of expression patterns in vivo must consider immune selection. We have now added a paragraph to discuss this topic to the Discussion section (see lines 505-517).

- l179 onwards: would it be possible to provide some information on the types of var genes being activated in either the 'single' or many 'state' (e.g. ups-type, chromosomal position); there has been previous work proposing switch rate differences and how these relate to var type, can this also be observed here?

As described in our responses to Reviewer 1, we have modified the figures that display var activation patterns to include ups-type (see annotation of ups type in the clones shown in Figure 1D, for example), and all the heatmaps now have a uniform arrangement of genes arranged by ups-type. This enables the reader to consider the characteristics mentioned by the reviewer.

- figure 4f: could these be put on the same scale?

We considered this, and recreated the graphs with a single scale, however the white space became quite large and we prefer the reduced scale to give the reader the most resolution of the data.

- figure 4 / 5 / general: it would be good to better differentiate the var2csa WT single and many lines to make it easier to see / understand that differences are also apparent in the wild-type lines (even though these will be smaller than for the var2csa-deleted ones).

We agree with the reviewer that the differences between wildtype singles and manys should be similar to the differences we observe between the var2csa mutant lines, albeit smaller. At the request of the reviewer, we now separate the wildtype “singles” and “manys” in Figure 5F. As predicted, this analysis shows the wildtype lines do in fact display intermediate levels of gene expression, consistent with a shift in cell cycle progression. We thank the reviewer for this excellent suggestion and agree that this is likely to make it easier for readers to understand this concept.

- l380: please clarify whether this implies that switch rates are variable / change over time, rather than being some fixed properties of different var genes, as previously proposed. Or is this with reference with on-switches, which would depend on previously expressed genes.

In this sentence, we were specifically referring to on-rates, rather than overall switch rates. We have modified the sentence to make this clear (see line 326).

Reviewer #2 (Significance (Required)):The authors provide seemingly robust evidence for both structured var gene switching in *P. falciparum* and the central role that var2csa plays in this process. Given that antigenic variation is paramount for immune evasion and acquired immunity and further influences malaria pathology, deepening our understanding of the molecular mechanisms underlying this process is highly important and should be of broad interest to a wide range of researchers.Reviewer #3 (Evidence, reproducibility and clarity (Required)):Zhang et al. aimed to gain further insight into the long-standing question of the mechanism of var gene switching in the malaria parasite Plasmodium falciparum. To this end, the authors first generated series of clonal parasite lines and report two different var gene expression states, "single" and "many", and were able to show that these additionally differ in the total amount of var transcripts. While this approach is not very innovative and has been used previously to study var gene expression patterns of *P. falciparum* isolates in vitro (e.g., also by the authors themselves Frank et al. 2007), the proposed two different states associated with differences in overall levels of var gene expression are novel and interesting. They linked this observation to a previously proposed model (Recker et al. 2011) based on in vitro var gene expression data of clonal lines, which implies a non-random, highly structured switch pathway in which an originally dominant transcript switches through a series of intermediates to either a new variant or back to the original. This so-called single-many-single pathway also applies to the in-host situation, in which an optimized switching network with sink and source nodes would allow the parasite to maintain and re-establish infections in humans. Because a previous publication by the authors suggests that var2csa may play a role in coordinating this var gene switching network, the authors used reverse genetics to render the var2csa locus inactive and examined its effects on var gene expression and switching. They demonstrated that (i) the var2csa locus is not required for gene expression or activation of other var gene variants, (ii) var2csa may be involved in var gene switching, as clonal var2csa KO lines show a remarkably stable var gene expression pattern over long in vitro culture periods, and (iii) var2csa KO lines show a shift in cell cycle progression estimated from RNAseq expression profiles. In addition, they analyzed a dataset from the malaria cell atlas (Howitt et al. 2019) to investigate whether the "single" and "many" states are also found at the single cell level, which would challenge the mechanism of mutually exclusive var gene expression. According to their analysis, in 15% of ring-stage parasites in which var expression was detected, reads were assigned to two different variants, suggesting that individual cells were in the "many" state, these also had reduced var expression, and that var gene expression is not always mutually exclusive.Major comments:(i) Clonal parasite lines show either "single" or "many" var gene expression profiles and can alternate between these two states.This data point is convincing because the authors analyzed many clonal cell lines (how many in total throughout the manuscript?)

We have added an additional dataset to Figure 1 which incorporates data from a large number of wildtype subclones. Figure 1D has been greatly expanded to include 41 closely related, clonal lines, and data included in a new Figure 1E includes data from a total of 74 lines.

Showing the existence of both states and the transition between them in the 3D7 parasite line used. However, why do the authors show only a single example of each state in Figure 1 without indicating how many clonal cell lines were made in total and how many clonal lines exhibited each state?

Figure 1 was intentionally kept simple to illustrate that the single and many states are reversible. However, given the reviewer’s comments, we have now expanded Figure 1D to show a larger set of wildtype subclones through multiple “generations”. We have also added quantification of the number of subclones displaying each phenotype and the level of var gene expression in the lines displaying either the “single” or “many” expression profile (Figure 1E).

The accompanying data on the difference in total var gene expression between the two stages is interesting, but needs further support from analysis of more clonal WT lines at the RNA and protein levels (biological replicates), which in turn would allow statistical analysis of more data points. And do the authors have any idea how long these states last?

We have added analysis of additional recently subcloned populations to figure 1D and E, including statistical analysis. We also analyzed several clones by RNA-seq in Figure 4F, and the trend is quite clear. Regarding how long these states last, this is shown in Figure 3A for wildtype parasites and in Figure 4A for var2csa deletion lines.

Also, it would be good to know if these states can be observed in other *P. falciparum* isolates or if this phenomenon is restricted to 3D7.

Most of the experiments described in the manuscript were performed in 3D7 due to the availability of tools to easily, rapidly and quantitatively examine the full var repertoire. In the absence of these tools, it is difficult to use the same approach in alternative isolates. However, Recker et al. also observed the single and many states in IT, a different genetic background, and found the pattern to be comparable to what they observed in NF54 and 3D7 (ref 22). This is where the concept of “single” and “many” comes from. We now explain more clearly that we are applying an optimized qRT-PCR method in 3D7 to investigate an observation originally made in IT (see lines 127-133). However, to ensure that our observations are not a result of exceptionally long-term culture of 3D7, we obtained a stock of NF54 (the original isolate from which 3D7 was derived) that has been cultured for many fewer generations and that maintains many wildtype qualities (knobs, sexual differentiation, etc). Cloning of this culture also yielded “single” and “many” lines with similar characteristics and frequency as our original experiments using 3D7. We now include these data in Supplementary Figure 1. We also now mention that two other labs have similarly observed dramatic changes in var expression levels and heterogeneity in both NF54 and IT (references 29 and 30), indicating that this phenomenon is not specific to our 3D7 line (see lines 151-157).

(ii) var2csa is the main "sink node" and required for var gene switching.The observation that many cell lines tend to express var2csa over time during in vitro culturing is well known, but also nicely illustrated in the data given here. However, as the authors also note, activation of var2csa would have to be unstable to meet the requirements of a sink node that allows the parasite to re-enter the "many" state and switch to another var variant. In contrast, however, var2csa expression in vitro appears to be quite stable, as also shown by the data in Figure 3A. Thus, it remains unclear how this could be achieved, and the authors only speculate that the "unique structure, including the presence of an upstream open reading frame and an untranslated mRNA" could lead to a switch to another variant. Do they think that in this intermediate state var2csa is not translated into a protein? And is the switching network from Recker et al. still working with a single dominant sink node or does it require multiple ones? Mechanistically, the switching of many parasites to var2csa expression might also be a consequence of the combination of a strong promoter that effectively maintains a euchromatin state (i.e., high levels of more stable H2Az/H2Bz nucleosomes that counteract heterochromatin) and frequent positioning within a locus that is poorly able to nucleate heterochromatin at that site (e.g., left chromosome 12). Consequently, when the subtelomeric sequences of var2csa are exchanged for those even less capable of propagating heterochromatin, its switching properties change and it dominates the transcriptome (shown in Duffy et al., 2009 and 2016). However, when the var2csa promoter is knocked out, parasites do not revert to normal switching with the next "strongest" var promoter, with the "weakest" neighboring heterochromatin nucleation taking over as the "switching" hub, so var2csa activation does indeed appear to be required for frequent var gene switching. The effects of other sequences in the vicinity of var2csa were not examined in this study, but it would be interesting to see what happens if the coding region of var2csa is replaced by another var gene variant, the var2csa promoter is replaced by a weak promoter (e.g., of var3 variants), or if only the TSS is mutated. Similar to the experiments of Duffy et al., did the authors try to enrich their clonal lines on a particular var gene (e.g., on ICAM-1 binding) to see if this still worked?

The reviewer mentions many clever experiments and hypotheses for further deciphering the mechanism by which var2csa serves as a node in the switching network. Many ideas mentioned here we have also considered and are designing experiments to test, although incorporating all of them into the current manuscript is not possible given time and space constraints. The concept of promoter competition, with var2csa being a strong promoter, is now described in the 2^nd^ paragraph of the Discussion section, and we added a discussion of the Duffy et papers from 2009 and 2016 (see lines 458-461). We did not enrich our populations for activation of specific var genes, however we noticed no differences when we examined populations expressing different dominant var genes.

(iii) Differences in the ring stage transcriptome explained by shift in cell cycle progression of "single" vs "many" parasites.Unfortunately, the rather short section on material and methods does not allow any conclusions to be drawn about how the synchronization of the parasite lines was actually performed. Since a number of different cell lines were used, the authors should explain how they ensured that they analyzed all lines at the same time point, i.e. at the same hour after invasion. Thus, it is not entirely clear whether the authors observed an altered transcriptional profile that can only be explained by the altered duration of the ring stage, or whether this observed difference in ring stage progression resulted in a generally different duration of IDC (do parasites emerge earlier in "many" stages than in "single" stages?). And can this be explained by the cost of investment in var mRNA and PfEMP1 protein?

We have greatly expanded the methods section to provide details into how we synchronized our parasite cultures to ensure that all were entering the replicative cycle simultaneously (see lines 589-598). We performed the experiments in the replication cycle immediately after synchronization, thus the differences in gene expression reflect how rapidly the ring stages progress, rather than the populations becoming out of sync with one another over time. We also find that the overall cycle progression is similar in all lines, which is also reflected by the timing of DNA replication (Figure 5G) and in the overall growth rates of the parasites (Supplementary Figure 2). These data suggest that the more rapid progression through the early stages of the replicative cycle does not slow parasite replication. The reviewer brings up the question of whether reduced var/PfEMP1 expression might result in faster progression through the cycle (as also mentioned by reviewer 1). While this is an interesting and plausible hypothesis, we don’t detect the expected change in replication rates when comparing “singles” vs “manys” (Supplementary Figure 2). We now describe this in the Discussion section (see lines 482-488).

Although the authors have already used two different approaches to estimate the "age" of parasites from RNAseq data (although the age window shown in figure 5C is very large), they could also use the approach of Tonkin-Hill et al. (2018), recently applied to several datasets, to determine the stage distribution as well as the mean hpi of parasites used for RNAseq analysis (Guillochon et al., 2022; Thomson-Luque et al., 2021; Wichers et al. 2021). In addition, it would be important to generate growth curves for the different cell lines with proper quantification of stages by manual counting or by using other available tools, which would allow to answer the remaining open questions and potentially support their conclusions at the cell biology level.

The reviewer is correct that we could use an additional method to estimate progression of the replicative cycle in each of the lines, although we find this unnecessary. The point we are trying to make in this analysis is that the “manys” progress more rapidly through the ring stage than the “singles” and that this difference explains the differences in the transcriptomes. Exactly at what point in the cycle the samples were obtained is less important. We feel the analysis we have included supports this conclusion well. Importantly, all 11 clonal populations fit this model precisely, providing additional confidence in this conclusion. We have now added growth curves (Supplementary Figure 2) as well as analysis of DNA replication by flow cytometry (Figure 5G) to further define the differences in cell cycle progression.

(iv) Individual parasites can occur in either "many" or "single" states, implying that a single cell can express multiple var gene variants, which in turn implies that the mechanism of mutually exclusive var gene expression is more plastic than previously thought.Apart from previous reports showing expression of two var genes in a single cell (Joergensen et al., 2010), the re-analysis of the scRNA-seq data by Howick et al. in its current state provides in my opinion no further support for leaky, mutually exclusive transcription due to the technical limitations of the 10x scRNAseq protocol used and could be omitted. Howick et al. used a 3' library prep kit (transcripts are sequenced from the 3' end), var genes are very similar in their 3' end coding for the ATS region, the Illumina reads are quite short (PE-75bp), and sequencing errors that occur could further complicate the correct assignment of reads to individual variants. 10x is also quite noisy and leaky, as indicated by the absence of var gene reads in 1/3 of the cells, and in the remaining cells the median var reads is only 4 (minimum: 1; maximum: 30). It is also already known from other multicopy gene families that RNAseq mapping requires further stringent adjustments to be variant-specific (Wichers et al., 2019). To show convincing support at the single cell level, the authors need to show the mapping of reads to different var gene variants in these 144 cells.

We agree with the reviewer that the data from the Howick paper are not ideal for the analysis we would like to perform. These limitations cannot be overcome with the current dataset, so we have taken the reviewer’s advice and omitted this analysis.

Should the authors qualify some of their claims as preliminary or speculative, or remove them altogether?The entire section on single cell transcriptomics is rather leaky and preliminary and should be removed or replaced by single cell data optimized for the detection of var gene expression, e.g. by covering the whole transcript.

As mentioned above, we have removed this section.

Overall, several parts from the results should be shortened and/or moved to the introduction or discussion (eg. line 111-122, line 156-164, line 181-193, line 277-287, line 499-507), which would also sharpen the manuscript. The comparisons to the olfactory receptor family in mammals is mentioned several times throughout the MS, instead of discussing the similarities and differences between both systems in a single paragraph.

We have reexamined the text with respect to the reviewer’s suggestions and made many changes. However, with respect to mentions of model systems like the olfactory receptor gene example, we prefer to keep the citations as is, because we think it helps to bring clarity to a very complex concept. Given that many readers are likely to not be experts in the concepts of mutually exclusive expression and transcriptional gene choice, we feel that giving ample examples and repeating them when necessary is helpful.

It remains unclear how var2csa fulfills the function of the most important sink node in the model proposed by Recker et al. since the model contains an entire network with multiple sink (and source) nodes. In terms of robustness and path length, does the model work with only a single dominant sink node? And how is inactivation of the relatively stable expressed var2csa achieved?

These are excellent questions. We would prefer to defer to mathematical modelers to determine how current models could be modified to accommodate a single, dominant sink node. This is not our expertise. How var2csa is inactivated is an active line of research ongoing in our lab. We suspect that the unique structure of the gene, including the uORF and its encoded protein, are likely involved, but this will require extensive additional work beyond the current manuscript to definitively determine.

Would additional experiments be essential to support the claims of the paper? Request additional experiments only where necessary to evaluate the paper as it is, and do not ask authors to open new lines of experimentation.- Cell biological assays (such as growth curves) should be performed to clarify the observed differences in cell cycle progression (see iii) above.

These have been performed and added to the manuscript (see Supplementary Figure 2) as well as flow cytometry measuring DNA replication (Figure 5G).

- As far as the reviewer understood, the authors investigated the Var gene expression pattern only in the long-standing culture-adapted 3D7 clone derived from the African isolate NF54. To rule out the possibility that the observed patterns are (in part) a consequence of longstanding in vitro culture selection or a specific feature of the 3D7 clone, the experiments to generate subclones with "single" and "many" status in genetically distinct parasite lines should be repeated, which would suggest that the claims of the paper are a general phenomenon of *P. falciparum*. This would, in my opinion, increase the relevance of the study.

The original identification of the single-many-single (SMS) model was from qRT-PCR and Northern blot data using the genetic isolates IT and 3D7 (Recker, 2011). In our paper, we attempted to verify and further examine this model using a somewhat refined quantitative qRT-PCR method. This employs a specific var primer set originally designed by Salanti et al., 2003 that has been modified and improved as the genome sequence of 3D7 has been updated. This is a unique primer set because it was developed to allow absolute quantification (see similar comments to reviewer 1). Unfortunately, this primer set is only suitable for analysis of 3D7 or NF54. We now explain this more clearly in the manuscript, noting that Recker et al. observed similar “many” and “single” expression patterns in the IT parasite line (see lines 127-135). We also note that Merrick et al. (2015) similarly observed stark differences in var expression levels using independently derived clones of NF54 and 3D7 (see lines 149-152), as did Janes et al. (2011) working with the IT line, indicating that this phenomenon is not unique to the lines we describe. Nonetheless, to ensure that our observations are not a result of exceptionally long-term culture of 3D7, we obtained a stock of NF54 (the original isolate from which 3D7 was derived) that has been cultured for many fewer generations and that maintains many wildtype qualities (knobs, sexual differentiation, etc). Cloning of this culture also yielded “single” and “many” lines with similar characteristics and frequency to our original observations using 3D7. We now include these data in Supplementary Figure 1.

- A limitation of the manuscript in its current form is the exclusive focus on the mRNA transcriptional level and the lack of protein level data. Do the observed differences at the var gene transcriptional level correlate with less PfEMP1 protein at the parasite bulk level? This would allow to determine the effects of "single" and "many" status on PfEMP1 levels and whether var2csa needs to be translated to act as a sink node, thus also providing insight into the mechanism, which would expand the scope of the study and potentially increase its relevance. If this is beyond the scope of this study, the authors should discuss these points in their revised manuscript.

We agree that expanding the study to PfEMP1 expression levels would be an excellent next step, and we intend to pursue these studies in the future. However, the current manuscript is focused on the var transcriptional network, coordination of var gene activation and the identification of var2csa as a potential sink node. In our view, these are important findings independent of protein expression levels. Nonetheless, we have added discussion of PfEMP1 expression to the revised Discussion section (see lines 482-488 and 505-517).

- Panning of var2csa deletion cell lines to a different receptor (e.g. ICAM-1) would clarify whether the cell lines are indeed irreversibly stuck on their previous var gene expression or whether switching with lower frequency still occurs.

The reviewer is correct that panning has the potential to determine if it is possible for var2csa-disrupted parasites to undergo switching, however we already have data indicating that this can occur at a low frequency. For example, in Figure 4 A and B, parasites expressing different var gene profiles can be observed, however as suggested by the reviewer, this appears to occur at a much lower frequency. We have modified the text to make this clear (see lines 295-298).

Are the suggested experiments realistic for the authors? It would help if you could add an estimated cost and time investment for substantial experiments.While the generation of subclones in genetically distinct parasite lines would require a considerable amount of time (for cultivation), the follow-up experiments themselves are neither technically demanding nor expensive, especially since genetically distinct parasite lines can be easily obtained from public sources (e.g. BEI Resources) or other malaria laboratories. Experiments on PfEMP1 protein levels can most easily be performed using commercially available anti-ATS antibodies. Performing the required cell biological assays should be a relatively quick and feasible experiment.

As mentioned above, the methodology we employed in this study is heavily reliant on the primer set of Salanti et al., which is only suitable for NF54 and 3D7. However, we now describe and cite published work from others that used both the NF54 and IT lines, and which displayed a similar phenomenon (lines 127-135, 151-154 and references 22, 29 and 30), and we have added a similar analysis from the original NF54 isolate that has not been in culture for as many generations as 3D7 (see supplementary Figure 1). We are unaware of any commercially available anti-ATS antibodies, despite trying to find them. Several past publications have described such antibodies, but to date we have been unable to acquire them. We will continue to explore alternative sources for such antibodies, however as mentioned above, we believe our focus on coordination of var gene switching and a description of the central role var2csa plays in the var gene transcriptional network is a significant contribution to our understanding of antigenic variation in malaria parasites. The cell biological assays (growth rate comparisons and cell cycle progression, supplementary Figure 2 and Figure 5G) have been performed as requested.

Are the data and the methods presented in such a way that they can be reproduced?The method section and some of the legends to the figures would benefit from a more detailed description that includes important details about the experimental setup, the period of in vitro cultivation of the parasites, etc. As mentioned earlier, it is only clear from the figures that the experiments were performed in the 3D7 parasite background. The synchronization procedures used in each experiment are not properly described, which seems particularly important in the context of the observed differences in cell cycle progression, and the origin of parasite lines "A3.B8" and "A3.C10" remains unclear.

We appreciate the reviewer’s comments on these details. We have greatly expanded the methods section and the figure legends to cover all the areas mentioned by the reviewer. We have also changed much of the annotation of the figures to make the experimental design and details clearer.

Are the experiments adequately replicated and statistical analysis adequate?- The analysis of total var gene expression levels shown in Figure 1C for the two subclones "C3C6" and "C3C2" should be expanded using the already generated var gene expression data of all subclones stratified for the "single" and "many" states to statistically determine whether the observed difference is a general feature of these two states.

At the reviewer’s request, we have added an analysis of these data to Figure 1E and found that the differences are consistent and highly statistically significant. It is also worth noting that a similar analysis was derived from RNA-seq (Figure 4F), which gave a similar conclusion as the qRT-PCR data shown in Figure 1C and E. Thus, we are confident in the conclusions.

- Figure 1B see minor comment below.Minor comments:Specific experimental issues that are easily addressable.- In the first Results section, the authors generated "several clonal parasite lines derived from a single parent population" that four weeks later showed two fundamentally different expression patterns. Figure 1 and the text show only two representative subclones (C3C6 and C3C2), which they claim to have used to verify the single-many-single theory, and which show a much lower overall level of var gene expression in the subclone with many expressed var genes. Since the authors have already produced more subclones, they should provide more replicates to support this. Is it a general feature of all "many" subclones to have lower total var expression (also applies to the data in Figure 2)? How many subclones were actually generated and how many showed which var expression phenotype?

We have added cumulative data from a large number (74) of recently derived subcloned populations to demonstrate that the level of var transcripts observed in distinctly and consistently different. These data are now shown in the greatly expanded clone tree in Figure 1D and in the analysis of total var expression in Figure 1E.

Do RNA expression levels correlate with protein levels?

As mentioned above, we do not currently have the ability to accurately assess protein expression levels but will attempt to do so in the future.

The authors could have used a different cell line to determine if this phenomenon applies to *P. falciparum* in general.

As mentioned above, the methodology we employed is limited to 3D7 and NF54, however we now cite other studies, including two that observed a similar phenomenon in the IT line (references 22 and 30).

- More details are needed for the experimental procedures:Only from the figures it is clear that the authors used the 3D7 parasite line for their study, which is not mentioned anywhere else.

Our choice of the 3D7 line is now explained in detail in the first section of the Results (see lines 127-134).

The procedure for obtaining synchronous parasites is insufficiently described with terms such as " highly synchronous cultures containing ring and trophozoite stage parasite were collected" (line 686ff). Is there a "or" missing here? Please describe exactly how you synchronize your parasites before performing qRT-PCR and RNAseq and provide the exact age of the parasites in hpi (+/-hpi). It is also unclear whether the parasite populations were synchronized immediately prior to RNA purification using Percoll/Sorbitol density gradient centrifugation, which at the very least has a massive impact on total RNA amounts and may also lead to variable quantitative qRT-PCR data.

We have added a greatly expanded Methods section to describe in detail how we synchronized our cultures (see lines 589-598). This includes “double synchronization” to obtain very tight synchronization. Importantly, parasites are allowed to recover from the synchronization procedure for 18 or 38 hours prior to RNA extraction, thus enabling the recovery of consistent amounts of RNA.

For qRT-PCR, it is unclear whether the entire run was always performed twice, as indicated by the statement in line 682f: "All qRT-PCR assays were performed at least in duplicates with no apparent differences between runs." It appears that the authors used the cycler (Quant Studio 6 Flex 489 realime PCR machine, Thermo Fisher) in a 96-well format and had to perform two runs per sample to cover all primer pairs in technical duplicates. However, since the cycler is also compatible with 384-well plates, it is not clear how they meet the standard of running all reactions in duplicate or even the recommended triplicate. Therefore, it is also unclear where the error bars in Figure 1B come from, which should be omitted if they come from technical duplicates, how much template was used for each qRT-PCR reaction and whether samples were checked for absence of genomic DNA before qRT-PCR or by using minus RT controls. In addition, the amplification efficiency of each primer pair is missing, skewing the comparison of transcript levels obtained with different primer pairs.

We have added details to the methods section to cover the questions raised by the reviewer. All qRT-PCR reactions were indeed performed in a 384-well format, enabling duplicates or triplicates to be performed together in a single run. We now explain this in detail in the figure legends and methods section (613-615). We have added details regarding controls for the presence of gDNA and how much template as used for each individual reaction (see lines 601-604). The PCR primer set employed here was originally created specifically to ensure similar amplification efficiencies for each primer pair and to enable absolute quantification (see Salanti et al., 2003). It has been used extensively by numerous groups when investigating var gene expression in the 3D7/NF54 genetic background. We have added this to the methods section (see lines 604-608). For Figure 1B, the error bars represent standard deviation from three biological replicates (see legend to Figure 1B).

Some more information should be provided for the RNAseq analysis. What was the actual RIN of the samples used for RNAseq? What library prep kit was used? How was the rRNA removed? How many cycles was the cDNA amplified during library prep, and was a polymerase suitable for high AT samples used? What was the actual fragment size, and were PE150 reads sequenced? What was the sequencing depth (total reads, mapped reads)? For analysis of gene families with multiple copies, mapping parameters may be critical to distinguish between variants (Wichers et al. 2019).

We have added the requested details to the Methods section (see lines 619-648), including an additional Supplementary table (Table S9). The analysis of gene families with multiple copies is not relevant since we omitted the multicopy gene families for the RNAseq analysis described in Figures 4 and 5.

- Please, use the term "rif" for genes and RIFIN for proteins. The used term "rifin" is a confusing combination.

We have corrected the text as requested.

- Figure 2A: Are the five new subclones derived from the same voluminous parasite culture as the first subclones?

As noted in our comments to reviewer 1, we have entirely renamed all the clonal lines to make it easier to determine their relationships to each other. We have also added a significantly larger number of clones to Figure 1D, including the 5 subclones shown in Figure 2A, to show exactly how the different parasite subclonal lines are related.

It is stated that these clonal lines were cultured in parallel for 70 generations and analyzed every 15-20 generations; however, 6 samples were shown for each clonal line, implying that the clones were cultured and analyzed for much longer than "just" 70 generations or were analyzed in shorter time periods. Please clarify and indicate the number of parasite generations at the top of the heatmap.

We have added the specific number of days in culture below each column of the heatmaps in Figures 2A and 4B to make the time of collection clear for each sample.

Hierarchical clustering should be shown on the left side of the heatmap. Is it correct that the gene PF3D7_0412700, which has a Z-score above 0 in all clones and at all time points, is not clustered with the genes showing the same pattern at the top of the heatmap!!!? Again, different groups of genes should be shown or color coded.

As also described in our response to reviewer 1, the heatmaps are no longer organized by hierarchical clustering. The genes are now arranged according to ups type.

Since all clonal lines, "manys" and "singles", show the same pattern of Z-scores, do they still differ in their total var gene expression level?

An extensive analysis of var gene expression levels is now provided in Figure 1E. Indeed, throughout all the clones analyzed (74 clones), the expression level is highly correlated with the many vs single phenotype (p<0.0001).

What is the origin of these other two clonal cell lines (A3.B8 and A3.C10)? There is no information about the origin or at least a reference. How do they know they are genetically identical, have they done whole genome sequencing?

These two clones have now been renamed “clone 4 and clone 5” to indicate that they are simply additional clones of 3D7 that were generated in our lab several years ago and cultured separately. They are thus derived from the same genetic background, but simply separated by time. This is now described in greater detail in the figure legend.

- Figure 2B&C: The term generation is somewhat misleading for the different "generations" of subclones, as it is also used for parasite generations in terms of the number of replication cycles of the parasite.

To avoid confusion, in the figure legend and the text we now refer to “subclone generations” and “replication cycles”. We hope this helps to avoid confusion.

Serial subcloning was performed only for clones in the "single" state, not for clones in the "many" state.

We have added a much more extensive subcloning tree in Figure 1D, including subcloning of both “singles” and “manys”. It is clear from this experiment that either phenotype can be generated from a population that displays the alternative phenotype. It is true that in Figure 2B, each round of subcloning originated from a “single” line, however Figure 2C includes heatmaps from both “singles” and “manys”, with no apparent difference.

How much time is between the actual cloning and the analysis? Please also include the parental 3D7 clone in the heatmap, put var gene group labels next to the IDs and labels for the different subclones so that the reader can easily see which pie chart from B belongs to which row in C.

The cloning process takes approximately 5 weeks from isolation of an individual parasite until a population has expanded sufficiently for the analysis to take place. We have added this to the methods section (see lines 574-577). The heatmaps are now organized according to var type. The rows in the heatmap in Figure 2C are now organized from left to right to match the pie charts in Figure 2B, thus enabling a reader to determine this relationship. We have added a description of this to the figure legend.

- As shown in supplemental Table S11, most of the 144 cells with two expressed var genes also expressed the var2csa variant and additionally another variant that was not mentioned in the manuscript but may be important because var2csa expression in the many-state is not detected as dominant by bulk RNAseq or qRT-PCR.

In response to a previous suggestion from this reviewer, we have deleted the single cell analysis because of the limitations of this dataset.

- The var2csa gene is expressed earlier than other var genes. Could this have an impact on the difference in cell cycle progression observed in the var2csa-deletion clones?

This is an interesting idea. We have no reason to think this might be the case, particularly since var2csa-mutant clones can be either singles or manys, and thus they can progress either rapidly or slowly through the ring stage. Without a better hypothesis, we have chosen not to discuss this idea.

- Which var genes are frequently found in minor transcripts and which in major expression states? Is there a correlation with the var gene group or position on the chromosome?

We have now organized all the heatmaps according to ups type and we don’t detect any correlation. However, these data are now available for readers to consider.

- The data of the cell line with the complete deletion of the telomere end of chromosome 12 shown in Figure 3B are not included in the manuscript, so this scheme and the mention of this cell line can be omitted.

We thank the reviewer for pointing out that we did not clearly describe the different methods for disrupting or deleting var2csa or how parasites derived from each method were analyzed. This is now shown in the revised Figure 4 as well as described in the revised text (see lines 260-271).

Are prior studies referenced appropriately?- Several parts of the introduction (eg. line 70ff.) that contain general statements about var genes and malaria pathogenesis would benefit from additional references.

We have added numerous references to this section to ensure the reader can easily access the relevant literature.

- Line 255: References for «Consistent with previous observations» seem to be missing.

We have added the appropriate references to this sentence (see line 243).

- The two papers by Duffy et al. (2009, 2016) that deal mechanistically with var2csa activation and control of switching are not mentioned at all, but the effects of the chromatin state of the var2csa locus should be discussed at least briefly.

We cite the Duffy papers in our discussion of var2csa promoter competition (see lines 458-461), where they fit nicely. We thank the reviewer for this suggestion.

- Correlation to in vivo var2csa gene expression data is lacking. For example, var2csa is also known to be expressed in children and males, suggesting a function other than "just" the receptor for CSA in pregnancy-associated malaria (Duffy et al., 2006; Beeson et al., 2007; Rovira-Vallbona et al., 2011). Or, Bachmann et al. (2019) have shown that an increase in var2csa expression parallels a change in var gene expression in a malaria-experienced volunteer infected with NF54.

We have incorporated these ideas and references into the Discussion section (see lines 475-478).

- An entire paragraph in the discussion addresses a case report of a splenectomized patient infected with non-adherent *P. falciparum* parasites that accordingly did not exhibit VSA expression. It is speculated that this may reflect the condition of "many" parasites with low var gene expression, however, it is rather unlikely that the qualitative approach in the cited study would not have detected these var gene transcripts, as they were also detectable by qPCR in the present study. Furthermore, this VSA-free phenotype can also be observed in other Plasmodium species that infect animals but lack var genes (e.g. Barnwell et al., 1983), suggesting that it may be a more general mechanism. However, this may indeed indicate that var gene expression could be regulated on a broader scale and in response to specific conditions.

This is an interesting point of discussion that we were hoping to stimulate with the paragraph in question. We note that indeed we were able to detect very low levels of transcript by qPCR in our study, however we were using specific primers optimized for the parasite line used in our experiments. In contrast, in the case of the splenectomized patient, the investigators had to use degenerate primers since the genome sequence was not known, which might have reduced sensitivity below the threshold of detection. In addition, the patient is presumed to have extensive anti-PfEMP1 immunity, thus providing a heavy selection pressure against *var* gene expression, a selection that is missing in our cultures. We now mention the similar example from *P. knowlesi* and have added the Barnwell reference (see lines 533-536 and the accompanying text).

Are the text and figures clear and accurate?- Figure 1: The histograms are too small and the transition groups might be placed between the corresponding main groups: A, B/A, B, B/C, C, E. The group D doesn't exist anymore, the var1 gene was included into the main group A (Kraemer et al., 2017 BMC Genomics). Dashed lines between the groups might help to see the classification of the expressed var gene on the first sight. It would also be useful to include the var gene group next to the annotation in the pie chart since on- and off rates might be influenced by the chromosomal location associated with the different var gene groups.

The text of the Figure has been enlarged to make it easier to read. We prefer to keep var1 as group D because this gene is unique amongst group A genes in that it does not recombine with other members of the family and represents an unusual gene that is conserved across species (see Gross et al., 2021). We have added the group designation next to the annotated pie charts as requested.

- Figure 3A: From PFD it not clear if all subclones tend to switch on var2csa over time, since the clones A9 and B6 seems to have different reddish color than the other clones. Since this is a general problem of multicolored pie charts, maybe indicate the percent of var2csa expression next to the corresponding pie?

We have replotted all of the pie charts to adjust the colors so that the red color that represents var2csa is conserved throughout the manuscript. Only var2csa is represented by red.

- Figure 4: The data shown in the different subpanels are not from the same clonal lines, which is not explicitly mentioned in the text. In addition, the data from the clonal lines should always be presented in the same clonal order.

To overcome the confusion from our original naming of the various clonal lines throughout the manuscript, we have renamed all clones in a systematic fashion. This also applies to the clones shown in Figure 4. This figure has also been reorganized to some extent for greater clarity.

Do you have suggestions that would help the authors improve the presentation of their data and conclusions?Figures would benefit from a more detailed labelling of the var gene groups, the time of cultivation of the different clonal lines and by showing the hierarchical clustering of var genes. Dominantly expressed variants should be clearly labelled in the pie charts.

As requested, we have added labeling of each var group and we indicate the dominantly expressed gene for each pie chart. We have also added the time of cultivation to each heatmap.

Reviewer #3 (Significance (Required)):Describe the nature and significance of the advance (e.g. conceptual, technical, clinical) for the field.Zhang et al. aimed to gain further insight into the long-standing question of the mechanism of var gene switching in the malaria parasite *Plasmodium falciparum*, and they are to be credited for tackling this important but experimentally difficult question. To this end, they sought to experimentally confirm earlier predictions of the var gene switching modeling studies of Recker at al. If successful, this would be a significant advance in the field. However, the reviewer is concerned that the manuscript in its present form provides data that, although partially consistent with the predicted models, are insufficient to be considered experimental proof of concept. Nevertheless, this study is not only of great interest to malaria researchers, but may also contribute to the understanding of similar mechanisms in other organisms. In addition, disruption of var gene expression and parasite antigenic variation may be a future target for intervention.Parasitologists, researchers working on host-parasite interactions or interested in gene regulation mechanisms might be interested in the reported findings.I'm working in the field of parasitology, malaria, var genes, but I'm not an expert in mathematical modeling.

[Editors' note: further revisions were suggested prior to acceptance, as described below.]

Recommendations for authors1. For many clonal lineages, only pie charts are provided for visual inspection and categorization into 'single' or 'many'. However, this also appears to be largely dependent on overall var gene expression, as in Figure S1A, subclone NF54-5 has dominant expression of a single variant and low overall var gene expression and was therefore classified as 'many'. The authors should provide expression levels as displayed in Figure 1C for all subclones analyzed in the study. These graphs could be displayed next to the corresponding pie chart, or as supplemental figures.Importantly, an unbiased approach to categorizing parasites into different states would be desirable. Perhaps the combination of a diversity index and total var gene expression could provide sufficient discrimination.

To address this suggestion, as well as comment 4D below, we have assembled a complete clone tree that includes all 74 lines derived from the subcloning experiments. This is quite large and perhaps a bit cumbersome to view, therefore we have included it as Figure 1—figure supplement 1. To make this large tree easier to understand, we modified the numbering scheme slightly. Two clones that were not derived from the subcloning experiments but were used for comparison in Figure 2A are also displayed here. We prefer to keep the smaller clone trees in the main manuscript (Figures 1D and 2B) where they are easier to follow and illustrate specific points. We included stacked graphs of total var expression for each clone in the clone tree in Figure 1—figure supplement 1 as requested.

The question of how to systematically categorize a subcloned population as either “single” or “many” is both challenging and somewhat arbitrary, as the reviewer correctly points out. We believe that no population is entirely “single” or “many”, but rather is a mix dominated by parasites in one or the other state. To test whether “single” vs “many” populations have different total var transcript levels as we do in Figure 1E, it is important that we not include transcript levels in the definition. Therefore, we now categorize populations in which more than 50% of the total var transcript is derived from 1 or 2 var genes as being “single” while populations with more diverse expression patterns are considered “many”. This stricter, unbiased definition resulted in only changing the classification of one population in the complete clone tree shown in Figure 1—figure supplement 1 (out of 74). We also changed the classification of the population specifically mentioned by the reviewer, NF54-5 in Figure 1—figure supplement 2. This population barely meets the definition of a “single” and has a correspondingly low transcript level. We now explicitly explain our methodology and our interpretation of the results on lines 172-182 and 192-198 of the main text.

2. The transcriptomics data reveal that "manys" progress through the first half of the replicative cycle faster, yet the "singles" catch up in the second half, and thus both populations have equivalent replication cycle lengths. The authors should discuss further why 'many' parasites have no growth advantage over several parasite generations, despite faster ring stage progression. Do the 'single' parasites catch up with this lag later? Giemsa smears, in addition to growth curves, would be useful to better document progression through the IDC of 'single' versus 'many' parasites.

Indeed the “singles” catch up with the “manys” and replicate at the same overall rate. This is reflected in three sets of data: (1) the RNAseq data (Figure 5A) show that at the trophozoite stage, there is very little difference in gene expression (compared to the substantial changes observed in the ring stages), indicating that the two populations are at nearly identical points in the replicative cycle, (2) analysis by flow cytometry (Figure 5G) shows that there is no detectable difference in DNA content between “singles” and “manys” (a reflection of nuclear replication), thus overall progression of the replicative cycle in not discernably different, and (3) the overall replication rate is not different when comparing the different populations (Figure 5—figure supplementary 1). We now make this point more clearly on lines 472-477. We could add Giemsa-stained smears, however we feel that the data we’ve already included are more quantitative and less subjective than smears and thus smears would not add to the manuscript.

3. The authors should discuss how the switch model proposed by Recker et al. could work with only a single dominant sink node that is barely inactivated during the cultivation of the parasites in vitro. In fact, what about the PF3D7_0421100 gene, which is also frequently activated and stably expressed in many subclones in different generations (Figure 2B)? Could this also be a sink node?

As mentioned previously, we are not mathematical modelers and would prefer to leave detailed discussions or the derivation of a new mathematical model to researchers who specialize in this methodology. We focused this manuscript on var2csa for the reasons detailed in the text, however we agree with the reviewer that it is possible that not all var genes are created equal. We have added to the discussion the possibility that other genes (and we specifically mention Pf3D7_0421100) could similarly serve as nodes in a hypothetical network (see lines 526-531). We thank the reviewer for this suggestion.

4. Reviewers appreciated that the authors have made an effort to improve the readability and presentation of the data. However, they also noted that there is still room for improvement.a) The labeling of the heatmaps and bar graphs is not consistent with respect to the order of var genes and the var groups are labeled twice in the heatmaps (largely on the left side and after each gene).

The labeling of the heatmaps have been modified to remove the redundancy of the var types, as suggested by the reviewer. We have also changed the order of the genes in the figures so that they all match.

b) There are some inconsistencies: cultivation days for the same clonal lines are not identical in Figures 2A and 3A;

We thank the reviewer for examining the figures so closely. There was indeed an error in assembling Figure 2A. We have corrected the time points and it is now clear that the times correlate between Figures 2A and 3A, although more time points are included in Figure 2. A later timepoint was included for clone 6 in Figure 3A because this more clearly shows convergence to var2csa expression. We now note that the time to convergence to var2csa is variable (see line 277), but all populations eventually express var2csa.

clonal line V2dis2 is classified as "many" in Figure 4A but as "single" in Figures 4E and F.

Once again, we thank the reviewer for catching this error. We now recognize that the population originally included in this figure (similar to the NF54 population discussed in response to point 1 above) was a poor choice given its initial expression profile, leading to confusion and mislabeling. To avoid confusion, we have replaced this population with an equivalent subclone in Figures 4A, 4E and 4F. This population was originally labeled V2dis3 and was included in the analysis in Figures 4E and 4F. We have now relabeled this as V2dis2 and show the pie chart for its var expression profile in Figure 4A. The population originally shown in Figure 4A is now labeled V2dis3 and is included in the analysis in Figures 4E and F. As expected, this population is the least polarized of the var2csa-mutant populations analyzed, although its var expression profile, as determined from RNA-seq, indicates it qualifies as a “single”.

c) Why do the authors not show var2csa expression in the wild-type heat maps, but in the pie charts?

The heat maps are designed to display the minor transcripts or “background” var expression pattern that we hypothesize might reflect switching biases. The heatmaps are presented so that readers can observe shifting patterns in the minor transcripts over time (Figure 2A) and when comparing closely related subclones (Figure 2C). var2csa is unique since it becomes activated in essentially all subcloned populations, thus we don’t consider it a “minor” transcript and it does not contribute to the purpose of these heatmaps. We now explain this in the text (see lines 228-230). We do indeed present expression data on var2csa, however we present it separately (see Figure 3A).

d) The phylogenetic trees of clones and subclones are partly redundant, but an overview including all clonal lineages, e.g. also clones 1, 2, 4, and 5, is still missing (could be included as a Supplemental figure).

As requested, we now present a complete tree with all related subclones in Figure 1—figure supplement 1.

e) Var1 should not be labeled as group D, which has been shown not to exist. The authors also do not explicitly designate var3 type genes, and the reviewers suggest including var1 and var3 in group A and more accurately labeling them var1 and var3 variants.

As requested, we now explicitly label var1 and the var3 variants, and include them as a subset of the group A var genes in Figures 1, 2, and 4.

f) The similarity of the minor transcripts between Figure 2A and C is difficult to judge from the heat maps. Why did the authors not include the subclones from Figure 2C in the Bray-Curtis dissimilarity analysis shown in Figures 4C and D?

The point of Figures 4C and D is to examine the differences between wildtype and var2csa-mutant lines, specifically by comparing how expression patterns shift over time. The data displayed in Figure 2C, while similar, does not reflect how the var pattern shifts within a population overtime, but rather how it shifts as populations go through the sequential bottlenecks of repeated subcloning. The analysis in Figures 4C and D thus reflects directly comparable experiments, while the data shown in Figure 2C is from a different experimental design. We now explain the nature of this comparison on lines 356-357. We are concerned that incorporating the additional, somewhat different experiment could be misleading. Further, adding another 5 sets of data points (44 additional dots) could obscure the message that we are trying to relate. We have therefore decided to leave the figure unchanged.

5. After disruption of the var2csa promoter, do the authors still see var2csa expression, or can they confirm its absence?

We checked for var2csa expression after disruption of the promoter using both Q-RT-PCR and RNA-seq and failed to detect any var2csa transcripts. We have added this confirmation to the text of the manuscript (see lines 303-305).